# Deep representation learning of chemical-induced transcriptional profile for phenotype-based drug discovery

Xiaochu Tong[1,2], Ning Qu[1,2], Xiangtai Kong[1,2], Shengkun Ni[1,2], Jingyi Zhou[1,3,4], Kun Wang[1,5], Lehan Zhang[1,2], Yiming Wen[1,2,6], Jiangshan Shi[1,2], Sulin Zhang [1,2] ✉, Xutong Li [1,2] ✉ & Mingyue Zheng [1,2,6] ✉

Artificial intelligence transforms drug discovery, with phenotype-based approaches emerging as a promising alternative to target-based methods, overcoming limitations like lack of well-defined targets. While chemical-induced transcriptional profiles offer a comprehensive view of drug mechanisms, inherent noise often obscures the true signal, hindering their potential for meaningful insights. Here, we highlight the development of TranSiGen, a deep generative model employing self-supervised representation learning. TranSiGen analyzes basal cell gene expression and molecular structures to reconstruct chemical-induced transcriptional profiles with high accuracy. By capturing both cellular and compound information, TranSiGen-derived representations demonstrate efficacy in diverse downstream tasks like ligand-based virtual screening, drug response prediction, and phenotype-based drug repurposing. Notably, in vitro validation of TranSiGen's application in pancreatic cancer drug discovery highlights its potential for identifying effective compounds. We envisage that integrating TranSiGen into the drug discovery and mechanism research holds significant promise for advancing biomedicine.

The field of drug discovery is experiencing a paradigm shift driven by artificial intelligence (AI). While target-based approaches have long dominated the field, their limitations—including a lack of well-defined targets, off-target effects, and unsatisfactory therapeutic responses—have driven the rise of phenotype-based methods. These approaches focus on the comprehensive cellular response to drug candidates, offering a more holistic understanding of disease mechanisms and potentially revealing novel drug targets and therapeutic avenues.

Transcriptomics data analysis plays a crucial role in drug discovery and understanding disease mechanisms. By capturing the global gene expression landscape across diverse biological contexts, it offers rich insights into cellular and organismal states. High-throughput RNA sequencing (RNA-seq) technologies have facilitated the generation of large-scale perturbational gene expression profiles, exemplified by databases like Connectivity Map (CMap)[1], the Library of Integrated Network-based Cell-Signature (LINCS)[2], PANACEA[3], ARCHS4[4] and ChemPert[5]. These large-scale perturbational gene expression profiles provide invaluable insights into how cells respond to various disruptions. The exploration of these profiles plays a central role in drug discovery, helping to elucidate mechanisms of action (MOA)[1,6,7]. For example, the CMap project proposed a pattern-matching strategy to identify compounds with shared MOA[1].

[1]Drug Discovery and Design Center, State Key Laboratory of Drug Research, Shanghai Institute of Materia Medica, Chinese Academy of Sciences, 555 Zuchongzhi Road, Shanghai 201203, China. [2]University of Chinese Academy of Sciences, No. 19A Yuquan Road, Beijing 100049, China. [3]School of Physical Science and Technology, ShanghaiTech University, Shanghai 201210, China. [4]Lingang Laboratory, Shanghai 200031, China. [5]School of Life Sciences, Division of Life Sciences and Medicine, University of Science and Technology of China, Hefei 230026, China. [6]School of Pharmaceutical Science and Technology, Hangzhou Institute for Advanced Study, University of Chinese Academy of Sciences, Hangzhou 310024, China. ✉e-mail: slzhang@simm.ac.cn; lixutong@simm.ac.cn; myzheng@simm.ac.cn

Furthermore, machine learning models like our proposed SSGCN can analyze relationships between chemical-induced and gene knockdown-induced profiles, offering a powerful method for identifying potential drug targets[8,9]. Additionally, generative models, such as Pham et al.'s FAME, applied to these perturbational profiles enable phenotypic molecular design[10–12].

Despite the immense value of perturbational gene expression profiles, the combinatorial complexity of drug-like molecules and cell lines limits exhaustive exploration through high-throughput experiments. This challenge has spurred the development of deep learning models capable of predicting transcriptional profiles for novel chemicals using publicly available data. DLEPS is a deep neural network designed to predict gene expression responses to new chemicals without cell-type specificity[13]. Furthermore, DeepCE[14] and CIGER[15] utilize one-hot encoding to distinguish between cell types, learning from diverse perturbational profiles. MultiDCP uniquely extends this by incorporating cellular context to predict both context-dependent gene expression and cell viability[16], enabling context-specific predictions for novel cell lines.

However, supervised learning models that directly fit gene expression values may struggle to distinguish true perturbation signals from confounding factors and the inherent noise within expression profiles. Recent studies highlight the power of variational autoencoders (VAEs) in handling high-dimensional, noisy transcriptomics data[17,18]. To address the limitations of data and generate novel perturbational profiles, we propose Transcriptional Signatures Generator (TranSiGen), a VAE-based framework leverages self-supervised representation learning to denoise and reconstruct transcriptional profiles, enabling the inference of new perturbational profiles. TranSiGen simultaneously learns three key distributions: the basal profiles without perturbation, the chemical-induced perturbational profiles, and the mapping relationship between them. This self-supervised approach effectively mitigates noise in the data and uncovers the underlying perturbation signals. TranSiGen offers several key benefits. (1) Improved inference of transcriptional profiles: TranSiGen's superior performance in inferring basal profiles, chemical-perturbational profiles, and the corresponding differential expression genes (DEGs) was demonstrated by comparisons with baseline models. (2) Unified representation for cellular and compound features: TranSiGen's generated perturbational profiles effectively capture both cellular and compound features, as evidenced by visualization analysis differentiating cell lines and drugs' MOA. (3) Versatile applications in downstream tasks: TranSiGen-derived representations have proven effective in various tasks including ligand-based virtual screening, drug response prediction, and phenotype-based drug repurposing. Its application in screening compounds against pancreatic cancer, with subsequent in vitro validation and high hit rates, demonstrates the power of TranSiGen's phenotype-based approach for identifying potent compounds. Importantly, TranSiGen's integration into phenotype-based drug discovery pipelines has the potential to significantly improve efficiency and reduce costs.

## Results

### The overview of TranSiGen

TranSiGen is a VAE-based model that simultaneously learns three distributions: basal profiles without perturbation, perturbational profiles, and the mapping relationship between them. It utilizes a self-supervised representation learning strategy to mitigate noise effects in the transcriptional profile and uncover the signal of perturbation.

The transcriptional profiles used in the model are obtained from level 3 data of the newly released CMAP LINCS 2020 dataset[2,19]. These profiles consist of 978 measured landmark genes per profile. Specifically, basal profiles ($X_1$) represent control profiles treated with DMSO, while perturbational profiles ($X_2$) represent transcriptional profiles treated with compounds. For each plate, the DMSO-treated control profile from the same plate is selected as $X_1$, forming a paired $X_1$ - $X_2$. The dataset includes 219,650 $X_1$ - $X_2$ pairs for 8316 compounds across 164 cell lines. Since L1000 assays are typically conducted with three or more biological replicates, there may be multiple $X_1$ - $X_2$ pairs for a perturbation-cell combination in the dataset. To ensure only one $X_1$ - $X_2$ pair per perturbation on each cell line, the repeated $X_1$ and $X_2$ pairs were further processed using the moderated-Z weighted averages algorithm (MODZ). The processed data consists of transcriptional profiles for 8316 compounds on 164 cell lines, including 78,569 $X_1$ - $X_2$ pairs (Fig. 1a).

TranSiGen consists of two VAE models: one encodes basal profiles $X_1$, and another encodes perturbational profiles $X_2$ (Fig. 1b and Supplementary Fig. 1). It learns to map from $X_1$ and the perturbation representation to $X_2$, which is denoted as $X'_2$. During inference, TranSiGen generates $X'_2$ from the input $X_1$ and the perturbation representation (Fig. 1b), and finally obtains the inferred DEGs $\Delta X'$ of the compound, where $\Delta X' = X'_2 - X_1$. A complete list of the symbols and notations used here were summarized in Table 1.

In downstream applications, TranSiGen can generate perturbational profiles for numerous compounds, allowing exploration of a larger space that is not covered by training data. The perturbational representation derived from TranSiGen can be applied to ligand-based virtual screening, drug response prediction in cells, and phenotypic screening of candidate compounds for disease (Fig. 1c).

### TranSiGen enables effective learning for transcriptional profiling

In this study, TranSiGen was used to simultaneously fit the basal profile $X_1$ and the perturbational profile $X_2$. The model's performance in learning $X_1$, $X_2$ and the corresponding DEGs $\Delta X$ was evaluated individually, where $\Delta X = X_2 - X_1$. As shown in Fig. 2a, TranSiGen exhibits excellent performance in reconstructing $X_1$ and $X_2$, denoted as $\hat{X}_1$ and $\hat{X}_2$, with the Pearson's correlation coefficients (PCC) close to 1 (between $\hat{X}_1$ and $X_1$, between $\hat{X}_2$ and $X_2$). It also performs well in inferring $X'_2$, which is predicted by $X_1$ and compound representation. Compared to directly evaluating the performance of learning $X_1$ and $X_2$, the corresponding performance in fitting DEGs is slightly decreased, with the PCC 0.734 and 0.619 in reconstructing $\Delta\hat{X}$ (between $\Delta\hat{X}$ and $\Delta X$) and predicting $\Delta X'$ (between $\Delta X'$ and $\Delta X$), respectively. Additionally, the relationship between TranSiGen's performance and $X_1 \sim X_2$ correlation coefficient ($R^2$) was analyzed. As shown in Fig. 2b, the sample size of the profiles increases with $X_1 \sim X_2$ $R^2$, as well as the prediction performance for DEGs. For $X_1 \sim X_2$ $R^2 > 0.8$, there is a slight decrease in performance, possibly due to the perturbation effects being too subtle for the model to fully capture. Overall, the model has learned the meaningful mapping from $X_1$ and compound to $X_2$.

Furthermore, we evaluated the profiling capabilities of TranSiGen by analyzing its effectiveness in learning cellular and compound representations in $\Delta X'$. Figure 2c presents a visualization of dimensionality reduction for both experimental $\Delta X$ and TranSiGen-derived $\Delta X'$, with each point color-coded by cell type. In the case of $\Delta X$, there was some clustering of the same cells, but also significant mixing between different cell types. In contrast, TranSiGen-derived $\Delta X'$ exhibited clear clustering of same cells and sharper distinctions between different cell types. This suggests that the representation derived from TranSiGen can more effectively differentiate between various cell types compared to experimental profiling, which is subject to high level of noise. Moreover, for compounds like decitabine, hydroxyurea and fludarabine, the $\Delta X'$ for different cells are closely grouped together, indicating their similar perturbation effects across different cells, as they all directly induce cell death due to cytotoxicity[20]. In addition to cytotoxic compounds, we expect other compounds sharing the same MOA to display similar effects on transcriptional profiling. The correlation between compounds with the

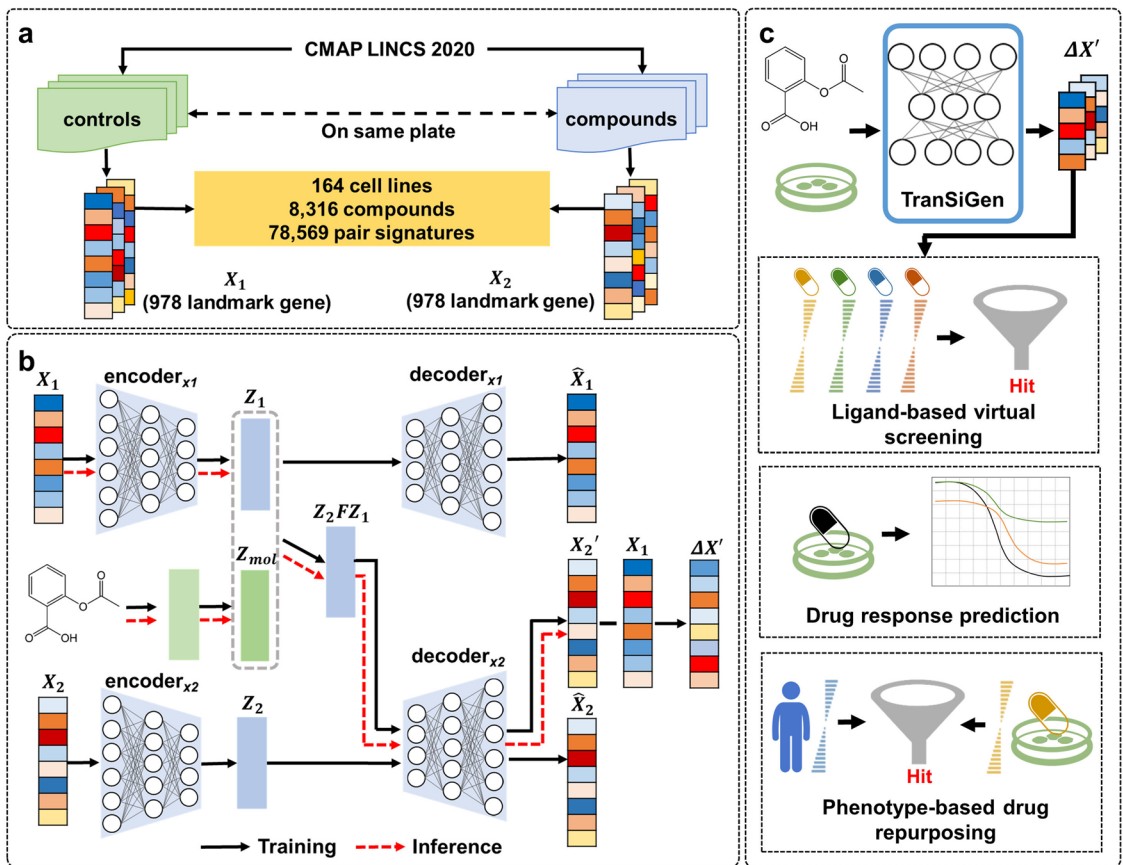

**Fig. 1 | TranSiGen's architecture and application. a** The data processing flow for TranSiGen. **b** The architecture and inference process of TranSiGen. **c** The applications of TranSiGen-derived representation. $X_1$ represents the control profile treated with DMSO, $X_2$ represents the transcriptional profile treated with the compound, $\hat{X}_1$ represents the reconstructed control profile, $\hat{X}_2$ represents the reconstructed transcriptional profile, $X_2'$ represents the predicted transcriptional profile, $\Delta X'$ represents the predicted differential expression profile, $Z_1$ represents the latent representation of $X_1$, $Z_2$ represents the latent representation of $X_2$, $Z_{mol}$ represents the hidden representation of the compound, $Z_2FZ_1$ represents the latent representation from $X_1$ and perturbation representation, $encoder_{x1}$ represents the encoder for $X_1$, $decoder_{x1}$ represents the decoder for $X_1$, $encoder_{x2}$ represents the encoder for $X_2$, and $decoder_{x2}$ represents the decoder for $X_2$.

## Table. 1 | List of symbols and notations used in the paper

| Symbol | Description |
|---|---|
| $X_1$ | The control profiles treated with DMSO |
| $X_2$ | The transcriptional profiles treated with compounds |
| $\Delta X$ | The differential expression genes $(X_2 - X_1)$ |
| $\hat{X}_1$ | The reconstructed control profiles |
| $\hat{X}_2$ | The reconstructed transcriptional profiles |
| $\Delta \hat{X}$ | The reconstructed differential expression genes $(\hat{X}_2 - X_1)$ |
| $X_2'$ | The predicted transcriptional profiles from $X_1$ and perturbation representation |
| $\Delta X'$ | The predicted differential expression genes $(X_2' - X_1)$ |
| $Z_1$ | The latent representation of $X_1$ |
| $Z_2$ | The latent representation of $X_2$ |
| $Z_2FZ_1$ | The latent representation from $X_1$ and perturbation representation |
| $c_{mol}$ | The input representation of the compounds |
| $z_{mol}$ | The hidden representation of the compounds |
| $encoder_{x1}$ | The encoder for $X_1$ |
| $decoder_{x1}$ | The decoder for $X_1$ |
| $encoder_{x2}$ | The encoder for $X_2$ |
| $decoder_{x2}$ | The decoder for $X_2$ |

same MOA was analyzed and is showed in Fig. 2d. TranSiGen-derived representations have higher PCC for compounds with the same MOA than $\Delta X$. Meanwhile, when compared to random MOA, TranSiGen-derived representations of the same MOA also exhibit relatively high PCC (Supplementary Fig. 2).

Overall, TranSiGen's self-supervised representation learning helps denoise and reconstruct transcriptional profiles, effectively identifying and learning meaningful cellular and compound representations from data.

### Comparison with existing models in inferring differential expression genes

This section evaluates TranSiGen's performance in predicting DEGs compared to established baseline models. We benchmarked TranSiGen against DLEPS[13], DeepCE[14], CIGER[15] and MultiDCP[16]. Notably, DLEPS, DeepCE, and CIGER primarily focus on de novo chemical profiling, with DeepCE and CIGER employing one-hot encoding for cell type distinction. In contrast, MultiDCP is the sole method among these baseline models that considers cellular context and specializes in predicting perturbational profiles for novel cell lines. Consequently, we assessed TranSiGen's performance in two settings:

(1) Chemical-blind splitting: this scenario includes two tests (Fig. 3a). In scenario 1-1, TranSiGen is compared to all models for its ability to predict DEGs of new compounds, using a dataset with 355 compounds across 7 cells to ensure comparability among models.

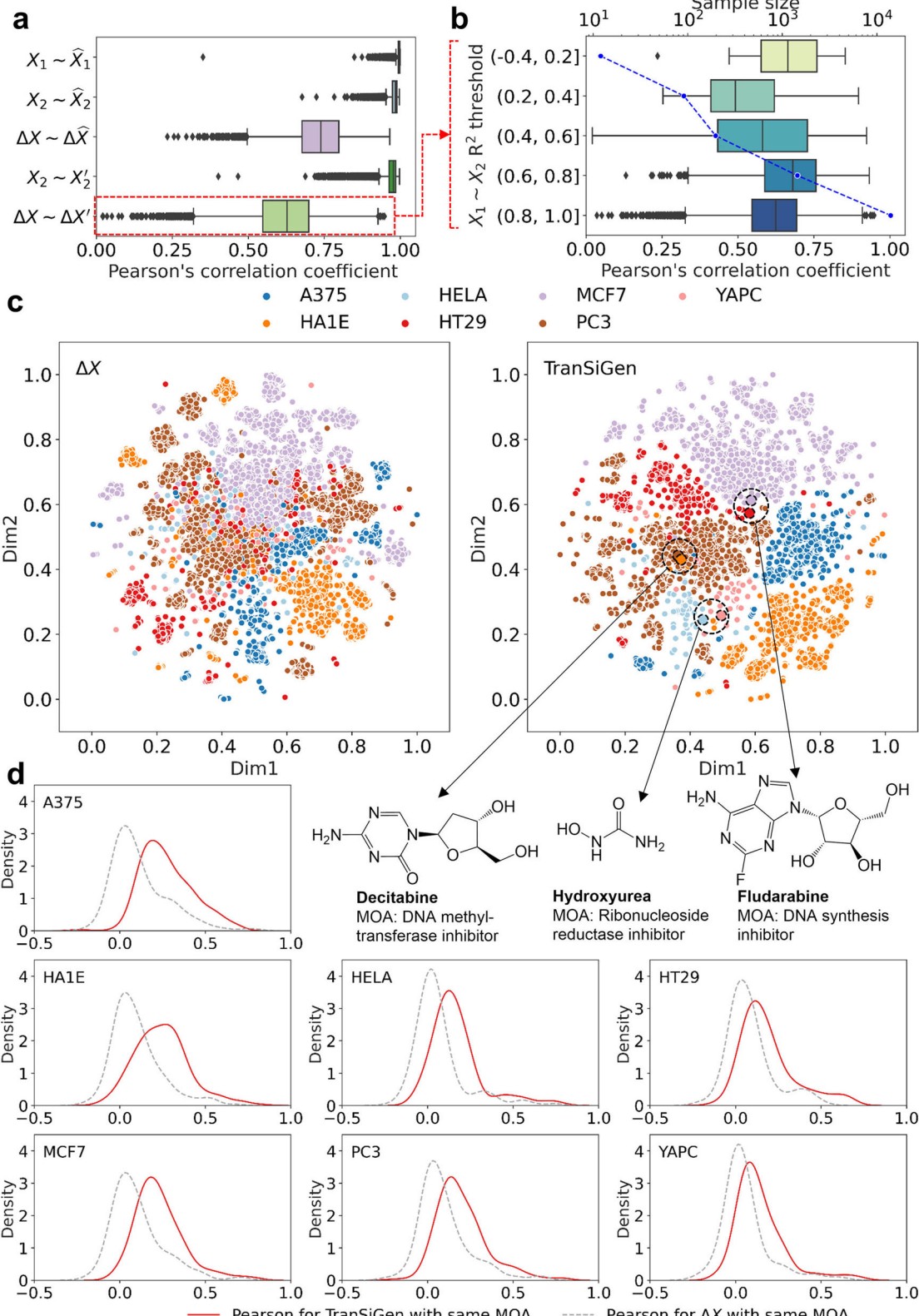

**Fig. 2 | Transcriptional profiling representation learning by TranSiGen.**
**a** Performance of TranSiGen in transcriptional profiling reconstruction and prediction (n = 15,713). Box-and-whisker plots show the median (center line), 25th, and 75th percentile (lower and upper boundary), with 1.5 × inter-quartile range indicated by whiskers and outliers shown as individual data points. **b** The change of TranSiGen's performance for $\Delta X'$ with the correlation between $X_1$ and $X_2$. Box-and-whisker plots show the median (center line), 25th, and 75th percentile (lower and upper boundary), with 1.5 × inter-quartile range indicated by whiskers and outliers shown as individual data points. The line plot corresponds to the sample size within each threshold, and the specific sample size is shown in source data. **c** Dimensionality reduction visualization using $\Delta X$ and TranSiGen-derived $\Delta X'$ for different cell lines. **d** Distribution of Pearson's correlation coefficients of profiles for the same MOA by $\Delta X$ and TranSiGen-derived $\Delta X'$. Source data are provided as a Source Data file.

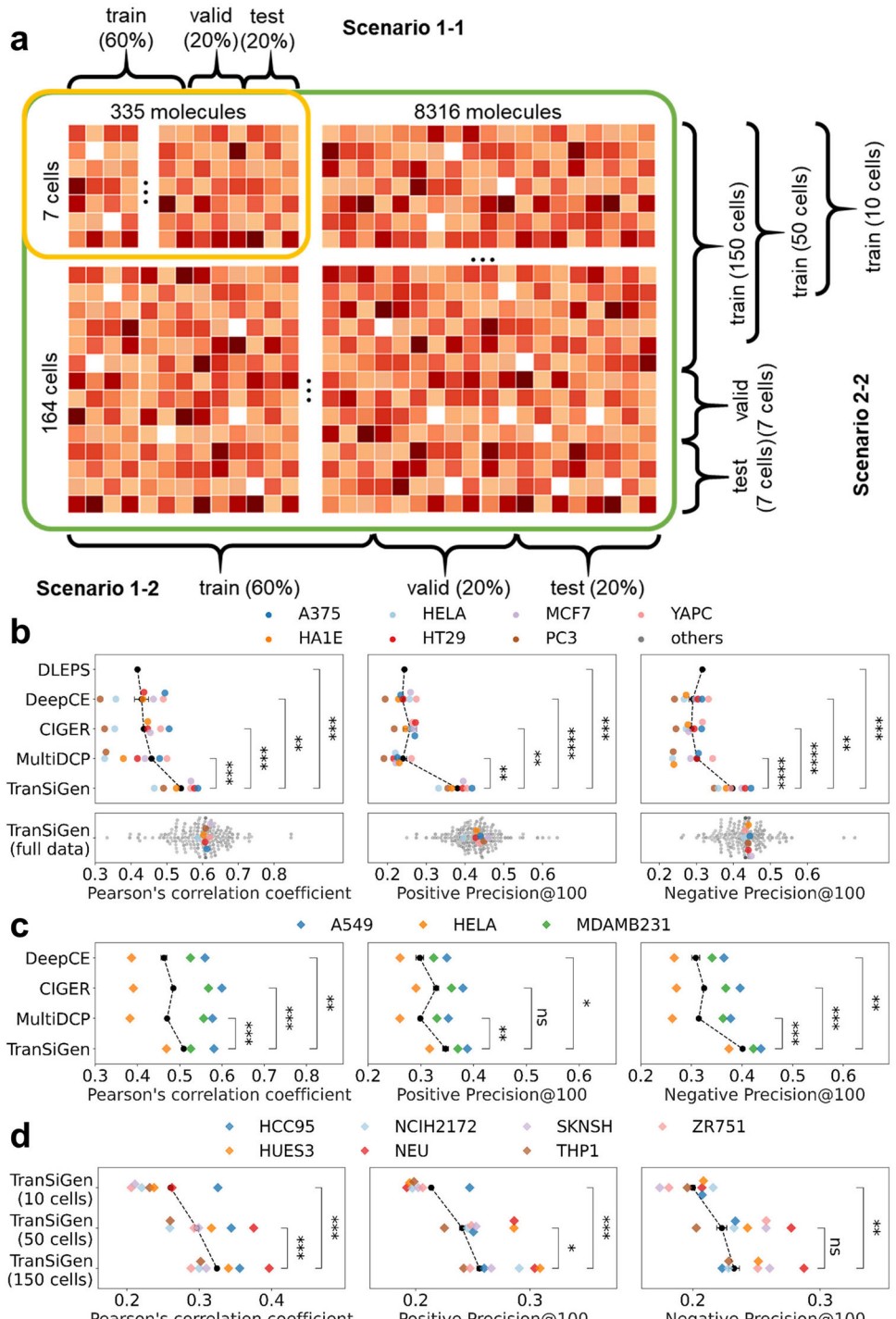

**Fig. 3 | The diagram of data splitting and the performance of inferring DEGs in different scenarios. a** The diagram of chemical-blind splitting and cell-blind splitting. In scenario 1-1, a dataset of 355 compounds on 7 cell lines is split by compounds, ensuring that test compounds do not seen in the training set. In scenario 1-2, a complete dataset of 8316 compounds on 164 cell lines is split by compounds. In scenario 2-2, the complete dataset of 8316 compounds on 164 cell lines is split by cell lines. The model was trained using the profiling data of 10, 50, and 150 cell lines, and the prediction performance was evaluated on 7 new cell lines. **b** Model performance comparison in chemical-blind splitting. **c** Model

performance comparison in cell-blind splitting (scenario 2-1). **d** The performance of TranSiGen in cell-blind splitting (scenario 2-2) by using different numbers of cell lines in the training set. All models were run three times with different random seeds. Black dots indicate the corresponding data points, and error bars represent the mean ± standard deviation. Two-sided *t*-test was applied between the models, and the exact *p* values are in source data. Source data are provided as a Source Data file. (****$p < 0.0001$; ***$0.0001 < p \le 0.001$; **$0.001 < p \le 0.01$; *$0.01 < p \le 0.05$ and ns, $0.05 < p \le 1.0$).

In scenario 1-2, the complete dataset with 8316 compounds across 164 cells is used to evaluate TranSiGen's scalability.

(2) Cell-blind splitting: this scenario also encompasses two tests. In scenario 2-1, we followed the challenging experimental setup proposed in MultiDCP[16], where testing cell lines significantly differ from the training set, to predict DEGs for new cell lines. TranSiGen is compared to all models, excluding DLEPS due to its inability to distinguish cell types. In scenario 2-2 (Fig. 3a), TranSiGen is trained on 10, 50, and 150 cells, then evaluated on 7 new cells, to assess the benefit of expanding training cell types.

Results for chemical-blind splitting (scenario 1) are shown in Fig. 3b. Detailed results, including time consumption, computational resources, and metric scores, are provided in Supplementary Tables 1–3. TranSiGen excels in predicting DEGs for unseen compounds (scenario 1-1, Fig. 3b top). Compared to other models (DLEPS, DeepCE, CIGER and MultiDCP), it achieves a higher average PCC across seven cell lines. Additionally, TranSiGen outperforms these models in Positive Precision@100 and Negative Precision@100, metrics focusing on the most significantly regulated genes. Notably, TranSiGen's computational cost remains comparable. Furthermore, training on the complete dataset (scenario 1-2, Fig. 3b bottom) yields state-of-the-art performance. This significant improvement across seven cell lines compared to scenario 1-1 (Fig. 3b top) highlights TranSiGen's ability to leverage more training data for accurate DEG inference.

Results for cell-blind splitting (scenario 2) are shown in Fig. 3c, d. Comprehensive cross-validation results in Supplementary Table 4. When comparing TranSiGen with other models (excluding DLEPS) using the challenging experimental setup from MultiDCP, TranSiGen consistently outperforms other models in average PCC, Positive Precision@100, and Negative Precision@100 (scenario 2-1, Fig. 3c). Furthermore, TranSiGen's performance in inferring DEGs for unseen cell lines improves as the number of training cells increases (scenario 2-2, Fig. 3d and Supplementary Table 5).

In addition, we further explored the impact of different molecular representations and model initialization methods in the context of chemical-blind splitting. Initializing TranSiGen with perturbational profiles generated by gene knockdown yields superior performance compared to random initialization. Additionally, using pre-training representation, Knowledge-guided Pre-training of Graph Transformer (KPGT)[21], further enhances the performance of inferring DEGs, surpassing the molecular fingerprint ECFP4 (as detailed in the Molecular representations in Method and corroborated by the metric scores in Supplementary Tables 2 and 3).

Overall, these analyses underscore the efficacy of TranSiGen's self-supervised representation learning approach for transcriptional profiling. TranSiGen surpasses all baseline models in predicting DEGs for unseen compounds (chemical-blind splitting) and unseen cell lines (cell-blind splitting). Notably, despite the challenges of cross-cell prediction due to combined cell type and state influence on transcriptional profiles, TranSiGen demonstrates superior generalizability across cell lines. This suggests that TranSiGen effectively leverages basal cell profiles, potentially mitigating the impact of cell type.

## Ligand-based virtual screening with TranSiGen-derived representation

Given that compounds with shared mechanisms induce similar gene expression profiles[1,7,22], we investigated the potential of predicted DEGs from TranSiGen as molecular representation for ligand-based virtual screening. First, as a proof-of-concept, Supplementary Fig. 3 demonstrates that active compounds targeting the same protein exhibit higher PCC compared to active and inactive ones. Subsequently, the TranSiGen-derived representation was used to assess whether a compound is active against a specific target. Specifically, we gathered and analyzed bioactivity data for compounds in LINCS 2020[19] and Pubchem[23] (refer to Ligand-based virtual screening in the

"Methods" section). We identified five distinct targets, namely HTR2A, DRD2, ADRA2A, SLC6A4, and KCNH2, each having a significant number of active compounds, as illustrated in Supplementary Fig. 4. For each target, random forest (RF) classifiers were trained to differentiate active and inactive compounds. Notably, we evaluated the performance of predicted DEGs from TranSiGen along with other baseline models in both chemical-blind and cell-blind settings for these screening models.

Figure 4a illustrates the performance of screening HTR2A (5-hydroxytryptamine receptor 2A) active compounds in chemical-blind setting, whereas Fig. 4b showcases the performance following the cell-blind setting. The model based on TranSiGen-derived representation outperforms other perturbational representations by a significant margin (Supplementary Tables 6 and 7). This result is further supported by the dimensionality reduction distribution of active/inactive compounds, where TranSiGen-derived representations clearly distinguish between the two, while other perturbational representations exhibit overlapped distributions (Fig. 4c and Supplementary Fig. 5).

Furthermore, leveraging TranSiGen's ability to capture compound characteristics across cellular contexts, we investigated whether fusing TranSiGen-derived representations from different cell lines improves compound screening performance. This analysis was conducted within the framework of a chemical-blind splitting scenario. Early fusion involves concatenating TranSiGen-derived representations from seven cells into one single feature, while late fusion merges the prediction results from seven cells. These two models are denoted as TranSiGen_EF and TranSiGen_LF for early and late fusion, respectively. It was observed that fusing TranSiGen-derived representations from different cell lines further enhances the screening performance of active compounds compared to individual cells alone (Fig. 4d). However, the performance improvement of TranSiGen_EF is not as significant as that of TranSiGen_LF, possibly due to the curse of dimensionality[24]. High-dimensional input features in TranSiGen_EF make it difficult to learn meaningful patterns. Similar phenomena are also observed in ligand-based virtual screening for other four targets evaluated (Supplementary Fig. 6).

As a molecular representation method, the TranSiGen-derived representation was compared to other molecular structural representations such as molecular fingerprint ECFP4 and the pre-trained representation KGPT. The maximum Tanimoto similarities of test molecules relative to training molecules were calculated using ECFP4. The performance of screening active compounds was evaluated at different maximal similarity thresholds. For compounds that are dissimilar to the training set (chemical structure similarity ∈ (0.0, 0.3]), the TranSiGen-based model demonstrates better predictive ability than structure representation-based model (Fig. 4e, Supplementary Fig. 6 and Table 8). This suggests that using transcriptional profiling, such as TranSiGen-derived representation, may have advantages in screening for new scaffold compounds that differ from known compound structures.

Therefore, TranSiGen-derived representation can be used as a new form of molecular representation for describing the characteristics of compounds from various cell contexts. It can also complement the structure-based representation and offer advantages in ligand-based virtual screening.

## Drug response prediction with TranSiGen-derived representation

Chemical-induced transcriptional profiles directly associate molecular features with the cellular effect of a particular drug. This association is beneficial for characterizing drug response in different cells[25–27]. Here, we applied the TranSiGen-derived representation to predict the area under the dose-response curve (AUC) of a compound on a specific cell line. The AUCs were obtained from the cancer treatment response portal (CTRP)[28,29]. We defined compounds with AUCs ≥5.5 as resistant

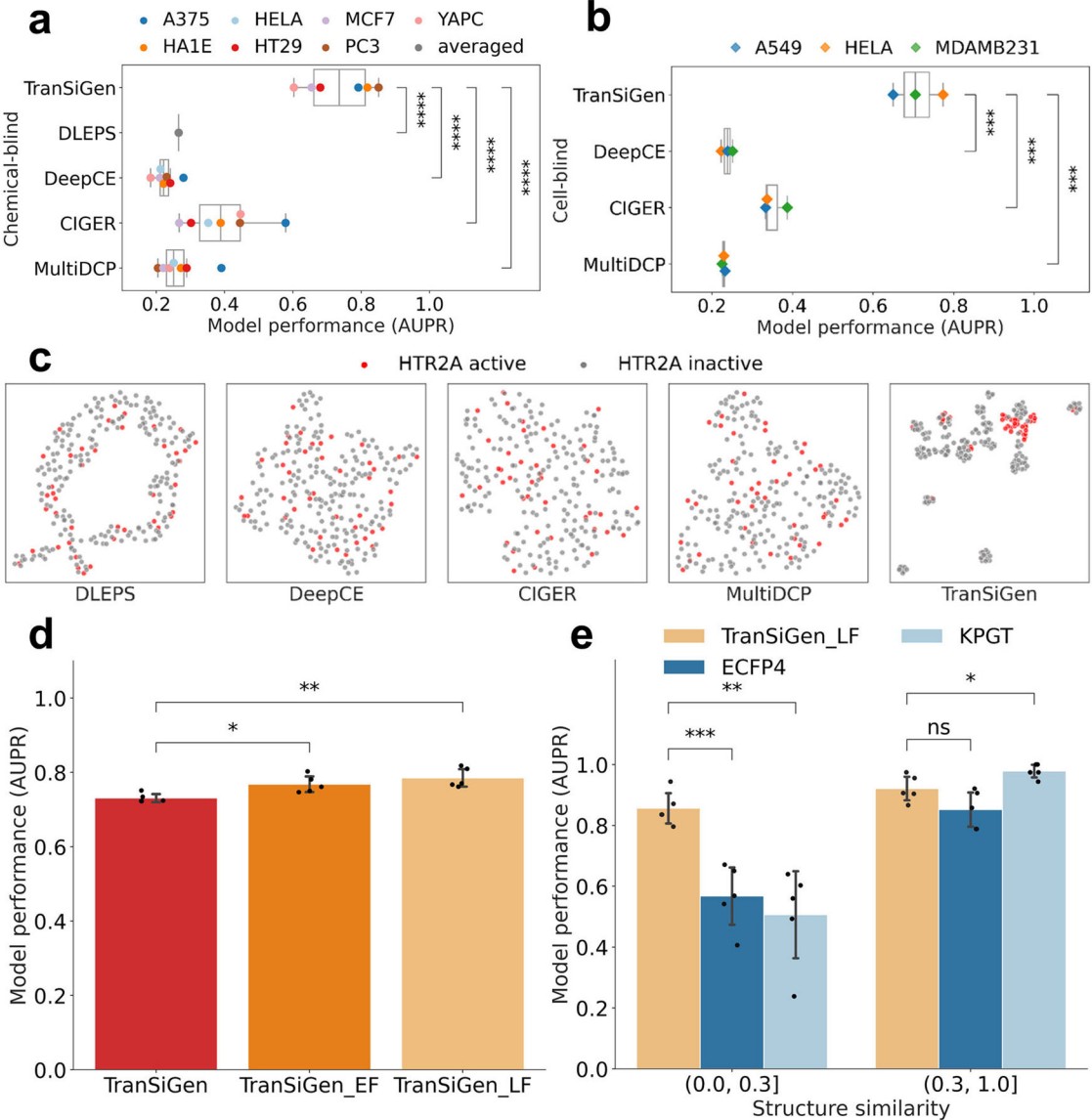

**Fig. 4 | Model performance of ligand-based virtual screening on target HTR2A.**
**a**, **b** Performance of active compound prediction using different perturbational representations in chemical-blind and cell-blind setting. Box-and-whisker plots show the median (center line), 25th, and 75th percentile (lower and upper boundary), with 1.5 × inter-quartile range indicated by whiskers. Colored dots indicate the corresponding data points for seven cell lines. Two-sided *t*-test was applied between the models, and the exact *p* values are in source data.
**c** Dimensionality reduction visualization of HTR2A active and inactive compounds based on various inferred perturbational representations. **d** Performance of active compound prediction by applying early fusion and late fusion for TranSiGen-

derived representation from seven different cell lines. All models were run five times with different random seeds. Error bars represent the mean ± standard deviation. Two-sided *t*-test was applied between the models, and the exact *p* values are in source data. **e** Performance of active compounds prediction within different thresholds of max similarity of test molecules relative to train data. All models were run five times with different random seeds. Black dots indicate the corresponding data points, and error bars represent the mean ± standard deviation. Two-sided *t*-test was applied between the models, and the exact *p* values are in source data. Source data are provided as a Source Data file. (****$p < 0.0001$; ***$0.0001 < p \leq 0.001$; **$0.001 < p \leq 0.01$; *$0.01 < p \leq 0.05$ and ns, $0.05 < p \leq 1.0$).

to cell lines, while those with AUCs <5.5 were considered sensitive[28]. More details about the dataset can be found in Supplementary Table 9.

To determine whether compounds can be classified as sensitive or resistant to a specific cell line based on TranSiGen-derived representation, we first assessed the profiling similarity among different compounds. In Fig. 5a, we calculated the PCC within a group of sensitive compounds (denoted as Sensitive), as well as the PCC between sensitive and resistant compounds (denoted as Sensitive-Resistant). Additionally, we compared the structural similarities of the two groups using Tanimoto similarity based on the molecular fingerprint ECFP4 (Fig. 5b). The results indicate that the Tanimoto similarities within Sensitive group and the Tanimoto similarities within

Sensitive-Resistant group are not significantly different on each cell line, suggesting that the structural representation ECFP4 cannot distinguish sensitive and resistant compounds (Fig. 5b). In contrast, we observed that the profiling similarities of Sensitive are significantly higher than those of Sensitive-Resistant on most cell lines (Fig. 5a). This finding demonstrates the effective discrimination between sensitive and resistant compounds achieved through TranSiGen-derived representation.

Furthermore, the TranSiGen-derived representation was used for drug response prediction in downstream task using a RF model. Its performance was compared with RF models based on other alternative representations, including perturbational representations generated

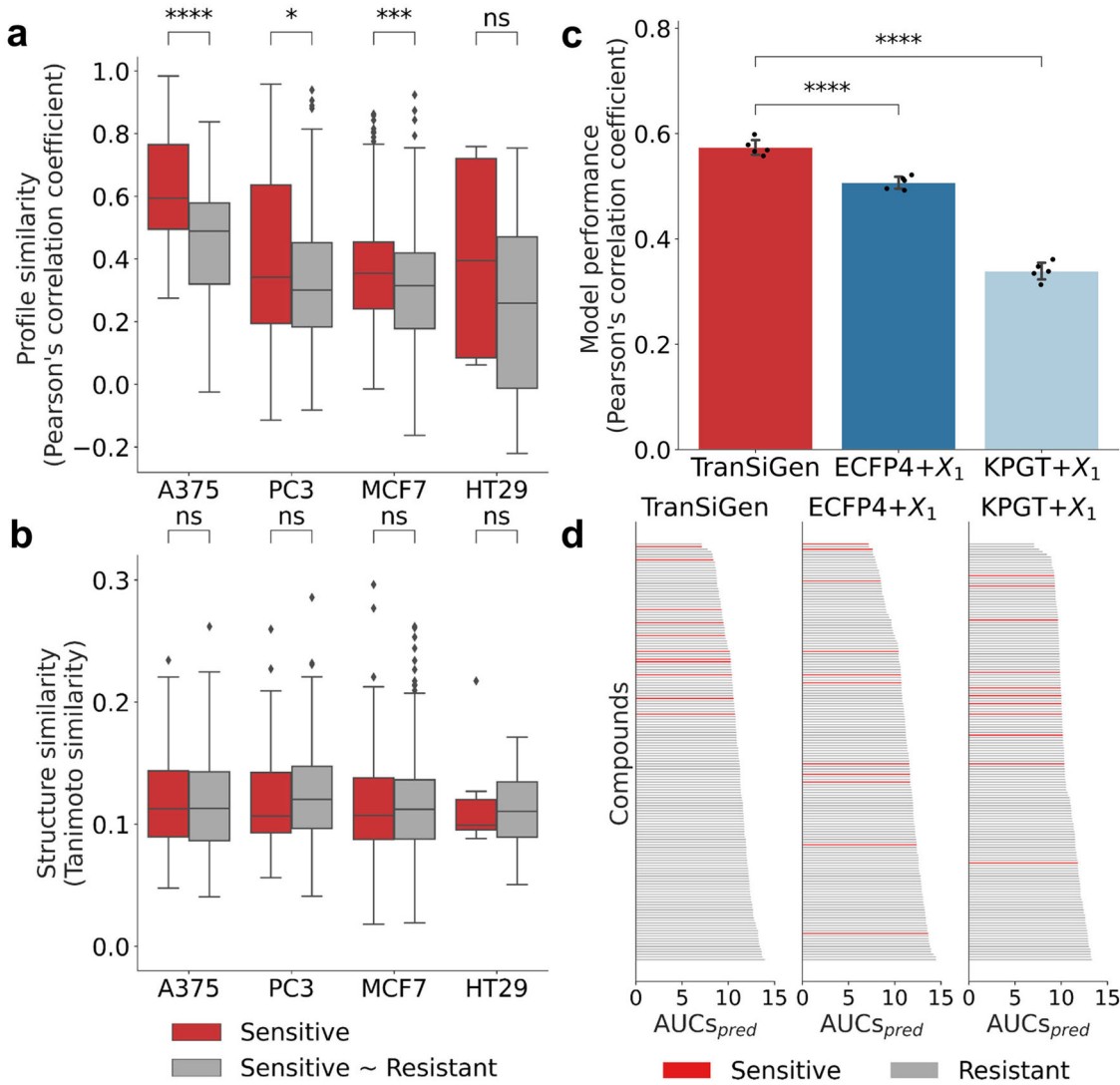

**Fig. 5 | Model performance of drug response prediction. a** The Pearson's correlation coefficients within a group of sensitive compounds and the Pearson's correlation coefficients between sensitive and resistant compounds based on TranSiGen-derived representation. Box-and-whisker plots show the median (center line), 25th, and 75th percentile (lower and upper boundary), with 1.5 × inter-quartile range indicated by whiskers and outliers shown as individual data points. The one-sided Mann–Whitney test was used to analyze the data. The exact *p* values and sample sizes are in source data. **b** The Tanimoto similarity within a group of sensitive compounds and the similarity between sensitive and resistant compounds based on molecular fingerprint ECFP4. Box-and-whisker plots show the median (center line), 25th, and 75th percentile (lower and upper boundary), with 1.5 × inter-

quartile range indicated by whiskers and outliers shown as individual data points. The one-sided Mann–Whitney test was used to analyze the data. The exact *p* values and sample sizes are in source data. **c** Performance of predicting drug response using various type of representations. All models were run five times with different random seeds. Black dots indicate the corresponding data points, and error bars represent the mean ± standard deviation. Two-sided *t*-test was applied between the models, and the exact *p* values are in source data. **d** Ranking results of compounds by AUCs$_{pred}$ of models based on various type of representations. Source data are provided as a Source Data file. (****$p < 0.0001$; ***$0.0001 < p \leq 0.001$; **$0.001 < p \leq 0.01$; *$0.01 < p \leq 0.05$ and ns, $0.05 < p \leq 1.0$).

by baseline models (DLEPS, DeepCE, CIGER and MultiDCP), as well as representations combining molecular structures and cell information (ECFP4 + $X_1$ and KPGT + $X_1$). As shown in Fig. 5c and Supplementary Fig. 7a, the TranSiGen-based model demonstrates significantly better performance than other models. Additionally, to evaluate the screening performance, compounds were ranked by their predicted AUCs (AUCs$_{pred}$), and classified as sensitive or resistant according to their true AUCs. The results showed that the TranSiGen-based model predicted sensitive compounds with smaller AUCs$_{pred}$ and higher rankings, while other models ranked the sensitive compounds randomly (Fig. 5d and Supplementary Fig. 7b). This indicates that the TranSiGen-based model has superior screening ability for sensitive compounds.

In summary, the TranSiGen-derived representation, simulating DEGs of compounds on cell lines, exhibits a distinguishable feature for

sensitive and resistant compounds and demonstrates remarkable performance on drug response prediction.

## Phenotype-based drug repurposing for the treatment of pancreatic cancer

Associating chemical-induced transcriptional profiles with diseases can help identify potential compounds for treating specific diseases[2,7]. TranSiGen-derived transcriptional profiles can be used alongside the profiles derived from chemical-treated and -untreated disease states to screen candidate compounds for disease treatment.

In this study, we integrated TranSiGen into a phenotype-based drug repurposing pipeline for pancreatic cancer[30] to assess its ability to prioritize sensitive compounds for the YAPC pancreatic cancer cell line from a pool of 1625 compounds in the PRISM Repurposing

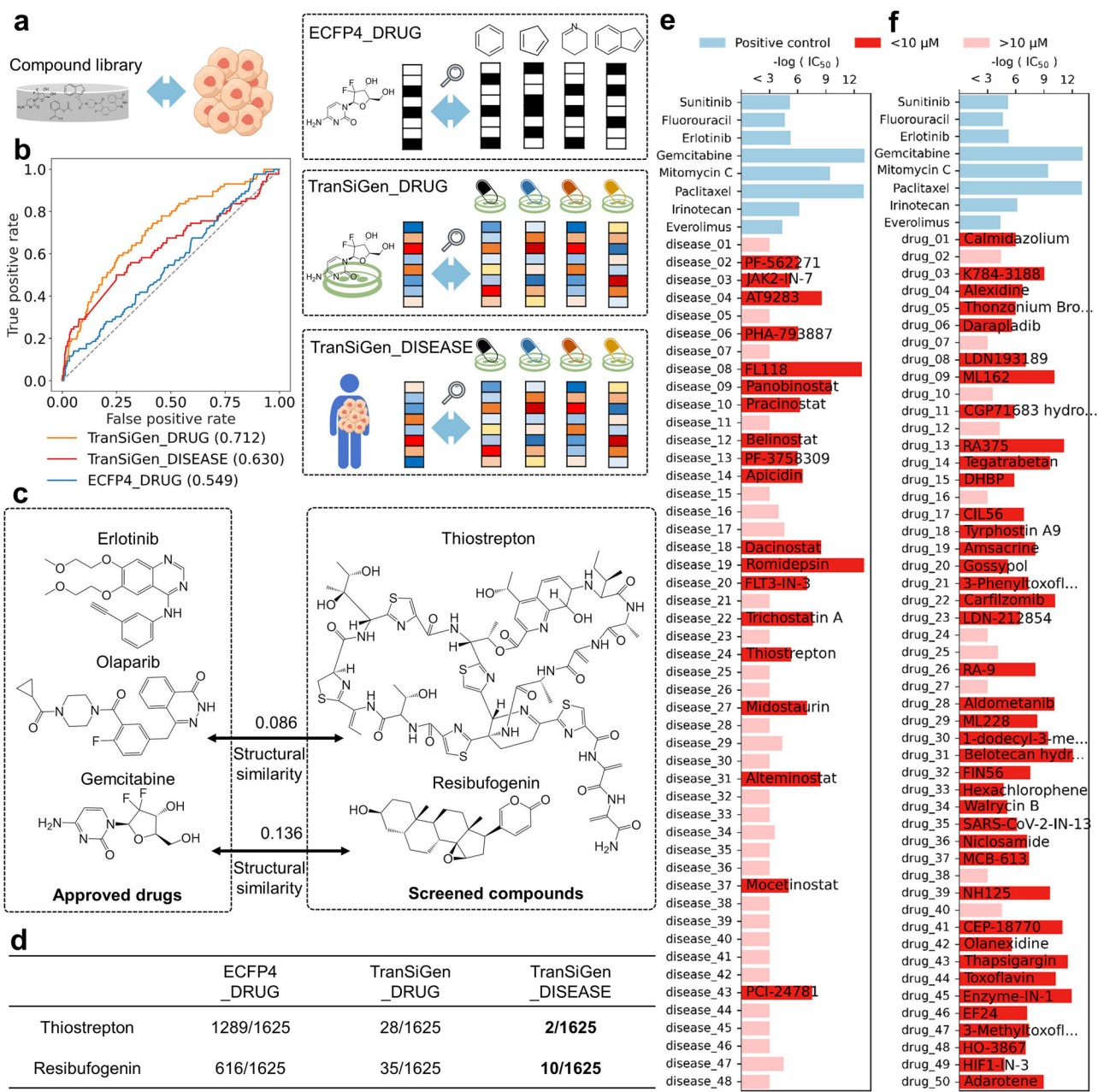

**Fig. 6 | Phenotype-based drug repurposing for the treatment of pancreatic cancer. a** The flow chart of drug repurposing strategy. **b** The screening performance of phenotype-based strategy and structural similarity-based strategy. **c** TranSiGen_DISEASE screened compounds that are capable of inhibiting pancreatic cancer cells, and their max structural similarities to approved drugs. **d** The rankings of thiostrepton and resibufogenin from different screening strategies. **e** Top 50 compounds screened by TranSiGen_DISEASE and their respective cell proliferation inhibition activities. **f** Top 50 compounds screened by TranSiGen_DRUG and their corresponding cell proliferation inhibition activities. Source data are provided as a Source Data file.

dataset[31]. We used two phenotype-based strategies and compared them to a conventional structural similarity-based protocol (Fig. 6a). TranSiGen_DRUG used the real DEGs of approved pancreatic cancer drugs to identify compounds with similar perturbation effects. Conversely, TranSiGen_DISEASE looked for compounds that can reverse the DEGs of pancreatic cancer. Both strategies used connectivity scores[32] to measure the relationship between DEGs. For comparison, ECFP4_DRUG was implemented to find compounds structurally similar to the approved drugs using ECFP4-based Tanimoto similarity. For more information, please refer to the section Phenotype-based drug repurposing for pancreatic cancer in the "Methods" section.

The screening performance of the three methods is shown in Fig. 6b. ECFP4_DRUG yields the worst predictive classification performance, follows by TranSiGen_DISEASE, and the best is TranSiGen_DRUG. Notably, the TranSiGen_DISEASE approach doesn't require any chemical-treated profiles, simulating scenarios where diseases lack known therapeutic drugs. This is a challenge not addressed by the structural similarity-based strategy. Even without using perturbed profiles of known drugs, TranSiGen_DISEASE effectively enriches hits among the top-ranking compounds (Supplementary Table 10). Phenotype-based strategies can identify compounds less similar to the approved drugs than those screened by ECFP4_DRUG (Fig. 6c and Supplementary Fig. 8). For instance, nature products thiostrepton and resibufogenin (Fig. 6c) ranks in the top 10 without sensitive annotations in PRISM dataset (Supplementary Table 11). Their abilities to inhibit pancreatic cancer cells have been confirmed by a literature

survey[33,34]. Thiostrepton, a natural cyclic oligopeptide, reduces the viability and clonogenicity of pancreatic cancer cell lines and induces ferroptosis via STAT3/GPX4 signaling[34]. Resibufogenin, a steroid lactone from the skin venom gland of toads, demonstrated potent anti-pancreatic cancer effects in vivo and in vitro, and can induce caspase-dependent apoptosis[33]. Figure 6d summarizes the rankings of these two compounds in different screening strategies. Both phenotype-based strategies, TranSiGen_DISEASE and TranSiGen_DRUG, consistently prioritizes them. In contrast, the structure-based strategy ECFP4_DRUG fails to effectively prioritize these nature products, ranking them at 1289 and 616, respectively. This may be attributed to the large structural differences between them and approved drugs, highlighting the inherent limitation of a structure-based strategy.

Moreover, we conducted a phenotype-based screening using the compound library available in our laboratory for pancreatic cancer and subsequently experimentally validated the top candidates in vitro. Specifically, we employed TranSiGen to predict the DEGs of 31,465 compounds on the YAPC pancreatic cancer cell line. These compounds were ranked using two distinct phenotype-based strategies, TranSiGen_DISEASE and TranSiGen_DRUG. Subsequently, we selected the top 50 compounds from each strategy to experimentally assess their activity on YAPC cells. Notably, positive molecules, including chemotherapy drugs (fluorouracil, gemcitabine, mitomycin C, paclitaxel and irinotecan), and targeted drugs (sunitinib, erlotinib and everolimus), were included as reference compounds (refer to the Experimental setting of validating screened compounds for pancreatic cancer).

Figure 6e, f displays the top 50 compounds identified through screening with TranSiGen_DISEASE and TranSiGen_DRUG, along with their respective cell proliferation inhibition activities. In total, the hit rates for TranSiGen_DISEASE and TranSiGen_DRUG are 38% and 80%, respectively. Here, a hit is defined as having an $IC_{50}$ less than 10 μM, a criterion comparable to that of the positive controls. The detailed prediction and experimental data for these top-ranking compounds are presented in Supplementary Tables 12 and 13.

Specifically, among the active molecules identified by TranSiGen_DISEASE, HDAC inhibitors (panobinostat, pracinostat, belinostat, apicidin, dacinostat, romidepsin, trichostatin A, alteminostat, mocetinostat and PCI-24781) consistently exhibit robust activities, indicating the promise of epigenomic therapeutics in pancreatic cancer[35]. On the other hand, the dominant category among the top 50 comprises selective kinase inhibitors, including a recently discovered JAK kinase selective inhibitor (JAK2-IN-7, identified in 2020), the aurora kinase inhibitor AT9283, and the FLT3 inhibitor FLT3-IN-3. Their kinase inhibitory activities may serve as the underlying mechanism for their anti-YAPC effects[36]. Among the active molecules identified by TranSiGen_DRUG screening, ferroptosis inducers exhibit robust activity, such as ML162, CIL56, and FIN56. This observation can be attributed to the crucial role of ferroptotic damage in both promoting and suppressing KRAS-driven pancreatic tumorigenesis[37]. Additionally, we examined two Wnt/β-catenin inhibitors, hexachlorophene and tegatrabetan. While hexachlorophene has been reported to reduce the proliferation of pancreatic cells[38], our newly identified compound tegatrabetan (64.04 nM) exhibits significantly higher activity compared to hexachlorophene (8.83 μM).

These results highlight the effectiveness of the phenotype-based strategies that use TranSiGen-derived representation in identifying potent candidate compounds, including those with unique structures. Overall, TranSiGen expands the range of compounds that can be screened with predicted perturbational profiles. It can be easily integrated into a phenotype-based drug repurposing pipeline, improving drug discovery efficiency and minimizing costs.

## Discussion

The field of drug discovery is undergoing a significant transformation with the integration of AI. While target-based approaches have been the dominant strategy, they are often hampered by limitations: (1) Many diseases lack clear and accessible protein targets, making it difficult to design effective drugs. (2) Targeting specific proteins can inadvertently interact with other molecules in the cell, leading to off-target effects and unexpected side effects. (3) Even when a target is identified, the resulting drug may struggle to reach its target within the cell due to poor cell permeability, thereby hindering the achievement of the desired therapeutic effect[39,40]. These challenges have spurred the emergence of phenotype-based approaches, which directly analyze overall cellular response to drugs, offering a more holistic understanding of disease mechanisms, and the potential for discoveries of novel drug mechanisms and therapeutic opportunities[13,41].

This study introduced TranSiGen, a VAE-based framework designed to address the limitations of supervised learning models for perturbational gene expression data. TranSiGen's novel self-supervised representation learning strategy effectively denoises transcriptional profiles. Extensive evaluations show that TranSiGen outperforms existing models in inferring basal profiles, chemical-induced perturbational profiles, and corresponding DEGs. This capability unlocks new avenues for expanding and enhancing existing drug discovery datasets. TranSiGen's core strength lies in its ability to overcome the noise and confounding factors inherent in gene expression profiles, offering a standardized way to characterize phenotypic information related to both cellular context and compound effects. This standardization facilitates integration and efficiency improvement across various downstream tasks, including ligand-based virtual screening, drug response prediction, and phenotype-based drug repurposing. Notably, its use in phenotype-based drug repurposing for pancreatic cancer, with subsequent in vitro validation, showcases its promise for real-world drug discovery scenarios.

TranSiGen sets the stage for continued exploration of VAE-based models and self-supervised learning approaches in drug discovery. Our future efforts will concentrate on addressing the heterogeneity of data from diverse sources in TranSiGen and enhancing the model's generalization performance concerning basal profiles from other platforms to broaden its application areas. Additionally, we plan to enhance the model's precision and interpretability by incorporating prior biological knowledge, such as pathways and gene ontologies. Beyond its current application in drug discovery, we are eager to investigate TranSiGen's potential utility in precision medicine and disease modeling, recognizing the substantial promise in these areas. The ultimate goal in this field is to create a truly comprehensive framework for efficiently utilizing high-dimensional gene expression data. This will accelerate drug discovery and unravel the complexities of disease mechanisms. TranSiGen, with its unique strengths and extensibility, marks a valuable step toward realizing this goal.

## Methods

### Transcriptional data processing

LINCS[2] has made publicly available resources on high-throughput gene expression profiles of different perturbations, such as small molecules and shRNAs. Using the L1000 assay, it is possible to measure the expression values of only 978 landmark genes, while still recovering most of the full transcriptome. The latest CMAP LINCS 2020 dataset was used in this study[19].

In LINCS, there are 5 levels of data. For this study, we utilized level 3 data, which included both raw perturbation profiles and control profiles (denoted as $X_1$ and $X_2$, respectively). To filter the profiles, we used the most common condition with a duration of 24 h and a dosage concentration of 10 μM for perturbed expression profiles by compounds. Additionally, we matched the expression profile with DMSO vehicle to the perturbed profiles on the same plate to create paired profiles $X_1$ - $X_2$, minimizing batch effects between cases and controls. The extracted dataset contained 219,650 $X_1$ - $X_2$ pairs for 8316 compounds on 164 cell lines. Furthermore, MODZ was applied to ensure

**Table. 2 | Description of the evaluation metrics**

| Evaluation metric | Equation[a] |
|---|---|
| RMSE | $\sqrt{\frac{1}{n}\sum_{i=1}^{n}(\Delta X_i - \Delta X_i')^2}$ |
| Pearson's correlation coefficient | $\frac{cov(\Delta X - \Delta X')}{\sigma_{\Delta X}\sigma_{\Delta X'}}$ |
| Positive Precision @100 | $\frac{G_{100-positive} \cap G'_{100-positive}}{G'_{100-positive}}$ |
| Negative Precision@100 | $\frac{G_{100-negative} \cap G'_{100-negative}}{G'_{100-negative}}$ |

[a] $n$ represents the number of landmark genes in expression profiles, $G$ represents the sets of top 100 positive/negative genes, $G'$ represents the sets of top 100 predicted genes.

that only one $X_1$ - $X_2$ pair per compound was included for each cell line with multiple $X_1$ - $X_2$ pairs. The processed dataset contained the transcriptional profiles of 8316 compounds on 164 cell lines, including 78,569 $X_1$ - $X_2$ pairs.

The gene expression profiles induced by shRNA were processed using the same method described above. Profiles from the 10 most common cell lines (A375, A549, ASC, HA1E, HCC515, HT29, MCF7, NPC, PC3, and VCAP) measured after 24 h were selected. The control profile with an empty vector in the same plate was then paired with the perturbed profiles. The final dataset contained 188,509 $X_1$ - $X_2$ pairs consisting of 4112 shRNAs on 10 cell lines, which was used to initialize two VAEs in TranSiGen.

## Molecular representations

Considering that the current number of compounds with experimentally measured gene expression profiles is still limited compared to the vast chemical space, TranSiGen utilized the pre-trained molecular representation KPGT[21] for compounds. KPGT is a novel self-supervised learning framework for molecular graph representation. It leverages a knowledge-guided pre-training strategy to capture rich structural and semantic information from large-scale unlabeled molecular graphs. In this study, the 2034-dimensional representation obtained from the KPGT pre-trained model was used as the molecular input for TranSiGen.

Alternatively, chemical fingerprints, are widely used as a form of molecular representation in machine learning, as they possess the virtues of being lightweight, computationally efficient, and ability to capture key molecular features[42]. Accordingly, we employed chemical fingerprints as the molecular input for TranSiGen. They are represented as binary vectors indicating the presence or absence of particular substructures in compounds. Specifically, the molecular fingerprint ECFP4[43] with a radius of 2 and a length of 2048 was used here.

## TranSiGen architecture

The VAE[44] is a deep generative model consisting of an encoder and a decoder. The encoder extracts significant information from the input, compressing it into a latent representation. Meanwhile, the decoder reconstructs a near-identical output from this latent vector. Consequently, VAE is capable of learning an efficient and meaningful latent space from high-dimensional data by compressing and reconstructing the original input. Unlike the standard autoencoder, which maps the input to a point in the latent space and trains by minimizing the reconstruction error, VAE encodes the input to a distribution. This requires the addition of a Kullback-Leibler (KL) divergence term to the reconstruction loss, which constrains the latent vectors to match a Gaussian distribution.

The architecture of TranSiGen consists of two VAEs: one for encoding the basal profiles $X_1$ and the other for encoding the perturbation profiles $X_2$. Each VAE comprises an encoder with two hidden layers ([1200, 100] dimensions) and a corresponding two-layer

decoder ([100, 800] dimensions). TranSiGen minimizes the loss of learning the representations of $X_1$ and $X_2$. Additionally, a linear function is used to map from the latent representation $Z_1$ of $X_1$ and the hidden representation $Z_{mol}$ of the compound representation $C_{mol}$ to the perturbed latent representation $Z_2FZ_1$ of $X_2'$, mimicking the chemical-induced transcription changes. The layer and dimension details of TranSiGen are shown in Supplementary Fig. 1.

During the training process, TranSiGen also minimizes the loss of predicting the differential expression genes $\Delta X'$. This involves minimizing the reconstruction loss between $X_2' - X_1$ and $X_2 - X_1$, as well as constraining the predicted perturbed latent representation $Z_2FZ_1$ match to the latent representation $Z_2$. The loss function of TranSiGen is defined as follow:

$$\text{Loss} = \text{MSE}(X_1,\hat{X}_1) + \text{MSE}(X_2,\hat{X}_2) + \text{MSE}(\Delta X, \Delta X')$$
$$+ \text{KL}(q(Z_1|X_1)||p(Z_1)) + \text{KL}(q(Z_2|X_2)||p(Z_2)) \quad (1)$$
$$+ \text{KL}(q(Z_2|X_1,C_{mol})||q(Z_2|X_2))$$

$$\text{KL}(q(Z_1|X_1)||p(Z_1)) = -\frac{1}{2}(1 + \log \sigma_1^2 - \sigma_1^2 - \mu_1^2) \quad (2)$$

$$\text{KL}(q(Z_2|X_2)||p(Z_2)) = -\frac{1}{2}(1 + \log \sigma_2^2 - \sigma_2^2 - \mu_2^2) \quad (3)$$

$$\text{KL}(q(Z_2|X_1,C_{mol})||q(Z_2|X_2)) = -\frac{1}{2}\left(1 + \log\frac{\sigma_2'^2}{\sigma_2^2} - \frac{\sigma_2'^2 + (\mu_2' - \mu_2)^2}{\sigma_2^2}\right) \quad (4)$$

where $\mu_1$ and $\sigma_1^2$ represent the mean and variance for $q(Z_1|X_1)$, $\mu_2$ and $\sigma_2^2$ represent the mean and variance for $q(Z_2|X_2)$, $\mu_2'$ and $\sigma_2'^2$ represent the mean and variance for $q(Z_2|X_1,C_{mol})$.

## Performance evaluation metrics

As shown in Table 2, the model's prediction performance was mainly evaluated using following metrics: Root mean squared error (RMSE), Pearson's correlation coefficient, and Precision@K. RMSE and Pearson coefficient were used to measure the prediction performance on the overall landmark genes. Precision@k, on the other hand, focused on the most significantly up- and down-regulated expressed genes. In this study, Positive Precision@100 was evaluated for the top 100 up-regulated genes, while Negative Precision@100 was evaluated for the top 100 down-regulated genes. We also evaluated additional regression metrics, including sum of squares due to error (SSE), mean square error (MSE), mean absolute error (MAE) and multiple $r^2$.

## Ligand-based virtual screening

We constructed a compound library (7148 compounds) for virtual screening against HTR2A. This library excluded the 355 compounds used for parallel performance comparison across seven cell lines, ensuring data integrity and preventing leakage. To assess model performance in downstream tasks, we predicted perturbation profiles for these 7148 compounds, which were then used to train the models. Target annotations for these molecules were obtained from two sources: the LINCS 2020[19] compound information file and PubChem[23] bioactivity data. For PubChem data, we considered compounds with $IC_{50}$, Ki, or $K_d$ values below 10 μM as targets. This process identified 41 active compounds for HTR2A. The remaining compounds in the external test set were randomly sampled at a 1:5 ratio to create a set of inactive compounds. Notably, similar screening was also conducted for four additional targets (DRD2, ADRA2A, SLC6A4, and KCNH2) using the same strategy as HTR2A. Finally, the compounds were split into training and test set with a ratio of 4:1 in each cell line.

Given the limited availability of dataset for active compound screening, we used RF for active compound prediction. To construct the RF classifiers, we used two types of features: inferred perturbational representations (TranSiGen, DLEPS, DeepCE, CIGER and MultiDCP) and structural representations (molecular fingerprint ECFP4, and pre-trained representation KPGT). We conducted a hyperparameter search for n_estimators, max_depth, criterion and obb_score to find the optimal model. To evaluate the model performance, we mainly used the area under the Precision–Recall curve (AUPR), and also calculated other classification-related metrics including the area under the receiver operator characteristic curve (AUROC), balanced accuracy (BACC), F-values (F1), Log_loss, and Matthews correlation coefficient (MCC). The training-evaluation procedure was repeated five times with different random seeds to determine the model performance. These processes were implemented using scikit-learn[45].

## Drug response prediction

The CTRP[28,29] is a widely used cancer cell response dataset that associate genetic, lineage, and other cellular molecular characteristics of cancer cell lines with drug sensitivity. It quantitatively profiles the sensitivity of cancer cell lines to small molecules. The AUC label is a dose-independent measure of compound sensitivity. Smaller AUCs indicate greater sensitivity of cells to the drugs. A subset of the drug response dataset for 267 compounds on four cell lines (A375, PC3, MCF7, and HT29) was obtained from CTRP, and the details of the dataset are shown in Supplementary Table 9. The processed dataset was split into training and test set at 4:1 ratio by compounds.

Similarly, RF regression models were used to predict drug response. Inferred perturbational representations (TranSiGen, DLEPS, DeepCE, CIGER and MultiDCP) and representations combining molecular structures and cell information ($ECFP4+X_1$ and $KPGT+X_1$) were used. Four hyperparameters, including n_estimators, max_depth, criterion and obb_score, were considered to obtain the optimal model. The model's performance was evaluated by the Pearson's correlation coefficient. The model's performance was assessed by repeating training-evaluation procedure five times with different random seeds.

## Phenotype-based drug repurposing for pancreatic cancer

**Differential gene expression profiles of approved drugs.** The approved drugs for pancreatic cancer were downloaded from https://www.cancer.gov/about-cancer/treatment/drugs/pancreatic. Among them, the DEGs of erlotinib, olaparib and gemcitabine were obtained from LINCS 2020 dataset[19]. These profiles were used for subsequent phenotype-based drug repurposing for pancreatic cancer.

**Differential gene expression profile of disease.** The pancreatic adenocarcinoma cohort of the The Cancer Genome Atlas (TCGA)[46] was downloaded from UCSC Xena (https://xenabrowser.net/). This cohort includes RNA-seq expression data of tumor samples and normal samples. The DESeq2[47] method was used to analyze the differential gene expression for pancreatic cancer. DEGs for pancreatic cancer were selected based on the following criteria: |log2Foldchange| > 1.5, $p$ value < 0.05 and false discovery rate < 0.25. A total of 293 up-regulated genes and 168 down-regulated genes were identified.

**Inferring perturbation gene expression profiles of compounds.** This study utilized the PRISM Repurposing dataset[31], which includes primary and secondary screening datasets, for phenotype-based repurposing for pancreatic cancer. The compounds from PRISM secondary screen were evaluated based on their AUC values, which indicate compound sensitivities on cells and serve as labels for screening performance assessment.

The dataset was downloaded from https://depmap.org/repurposing/. Compounds with ground-truth expression profiles in the LINCS 2020 dataset were excluded, resulting a dataset contains

1625 compounds. TranSiGen inferred the DEGs $\Delta X'$ of 978 landmark genes associated with these compounds in YAPC pancreatic cancer cell. Additionally, the expression values of 9196 best inferred genes were inferred from the generated 978 landmark genes to obtain the predicted expression values of 10,174 genes. The inference weight matrix was obtained from the L1000 project[2].

**Connectivity score.** The connectivity score, obtained from the gene set enrichment analysis[32], is used to measure the relationship between transcriptional profiles. The connectivity score ranges from −1 to 1, where −1 indicates a complete reversal of the query profile to the reference profile, while 1 indicates a complete similarity of the query and the reference profile.

Firstly, the enrichment score (ES) is used to evaluate the enrichment of a predefined gene set at the top or bottom of the reference differential gene list. The enrichment scores for up-regulated gene set and down-regulated gene set are denoted as $a$ and $b$, respectively:

$$a = \max_{j=1\,to\,t}\left[\frac{j}{t} - \frac{V(j)}{n}\right] \tag{5}$$

$$b = \max_{j=1\,to\,t}\left[\frac{V(j)}{n} - \frac{(j-1)}{t}\right] \tag{6}$$

$$ES = \begin{cases} a, if\ a > b \\ -b, if\ b > a \end{cases} \tag{7}$$

where $n$ represents the number of genes in the expression profiles, $t$ represents the number of genes in the predefined gene set, $V(j)$ represents the rank of a specific gene in the rank list, and $j$ represents the index of gene ranging from 1 to $t$.

Next, the above equations are used to calculate the enrichment scores of the predefined up-regulated and down-regulated genes by the query profile, resulting in $ES_{up}$ and $ES_{down}$. Finally, considering these two enrichment scores together, the connectivity score of the query profile relative to the reference profile is calculated as follows:

$$\text{Connectivity score} = \begin{cases} ES_{up} - ES_{down}, if\ sign(ES_{up}) \neq sign(ES_{down}) \\ 0, otherwise \end{cases} \tag{8}$$

Specifically, the reference profiles consist of the inferred DEGs of all compounds from TranSiGen. The DEGs of approved drugs for pancreatic cancer were used to identify compounds with positive connectivity scores, while the DEGs of pancreatic cancer were employed to identify compounds with negative connectivity scores. In the screening dataset, the top 20% compounds having the lowest AUCs on the YAPC pancreatic cancer cell line were identified as hits, and the area under the ROC curve was used to evaluate the screening performance.

## Experimental setting of validating screened compounds for pancreatic cancer

**Compound.** The 31,465 compounds screened against the YAPC pancreatic cancer cell line were mainly from the following sources: MedChemExpress (Monmouth Junction, NJ, USA), the ChemDiv Library (San Diego, CA, USA) and the Chemspace Library (Monmouth Junction, NJ, USA). Detailed information concerning the origins of the compound libraries and corresponding catalog numbers for the top 50 compounds identified through TranSiGen_DISEASE and TranSiGen_DRUG screening are presented in Supplementary Tables 12 and 13.

**Cell culture.** Human pancreatic cancer cell line YAPC was purchased from Cobioer Biosciences Co. Ltd (Nanjing, China). YAPC cells were

cultured in RPMI-1640 medium (BasalMedia, L210KJ) supplemented with 10% fetal bovine serum (FBS) (MeilunBio, PWL001) and incubated at 37 °C under a humidified, 5% (v/v) $CO_2$ atmosphere.

**Proliferation assays.** Cells were seeded in 384-well cell culture plate (NEST, 760601) at a density of 100 cells per well, and incubated with serially diluted compounds at concentrations ranging from 0.1 nM to 50 µM in a final volume of 50 µl. After 72 h, the inhibitory effects of test compounds on the proliferation ability of YAPC cells were determined using the CellTiter-Meiluncell Luminescent Cell Viability Assay Kit (MeilunBio, PWL111-3) following the manufacturer's instruction. Briefly, 25 µl of CellTiter-Meiluncell reagent was added to each well, and the plates were incubated on orbital shaker for 10 min at room temperature. Luminescence was measured on Perkin Elmer Envision multimode plate reader. $IC_{50}$ values were determined by nonlinear regression (curve fit) using a variable slope (four parameters) in Graphpad Prism (8.0.1).

### Statistics and reproducibility
A two-sided *t*-test was conducted for analyses related to model performance, while a one-sided Mann–Whitney test was utilized for analyses regarding transcriptional profiles. Detailed descriptions are provided in the figure legends. The significance level was set as ****$p < 0.0001$; ***$0.0001 < p \le 0.001$; **$0.001 < p \le 0.01$; *$0.01 < p \le 0.05$ and ns, $0.05 < p \le 1.0$.

### Reporting summary
Further information on research design is available in the Nature Portfolio Reporting Summary linked to this article.

## Data availability
All relevant data supporting the key findings of this study are available within the article and its Supplementary Information files. The raw expanded CMap LINCS Resource 2020 is available at https://clue.io/data/CMap2020#LINCS2020. The raw PRISM Repurposing dataset is available at https://depmap.org/repurposing/, and the raw pancreatic adenocarcinoma cohort of the TCGA is available at https://xenabrowser.net/. The analysis and results data generated in this study are provided in the Source Data file. All data are available from the corresponding author upon request. Source data are provided with this paper.

## Code availability
The code for model training and analysis is available at Github https://github.com/myzhengSIMM/TranSiGen and Zenodo https://zenodo.org/records/11435859[48].

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

## Acknowledgements

We gratefully acknowledge financial support from National Natural Science Foundation of China (T2225002, 82273855 to M.Z. and 82204278 to X.L.), SIMM-SHUTCM Traditional Chinese Medicine Innovation Joint Research Program (E2G805H to M.Z.), Shanghai Municipal Science and Technology Major Project, National Key Research and Development Program of China (2023YFC2305904 and 2022YFC3400504 to M.Z.), and The Youth Innovation Promotion Association CAS (2023296 to S.Z.). We thank the staff members of the Large-scale Protein Preparation System at the National Facility for Protein Science in Shanghai (NFPS), Shanghai Advanced Research Institute, Chinese Academy of Science, China for providing technical support and assistance in data collection and analysis.

## Author contributions

M.Z. and X.L. conceived the project and were responsible for the decision to submit the manuscript; S.Z. participated in guiding the implementation of the project; X.T. implemented the TranSiGen model, conducted computational analysis and wrote the paper; J.Z. completed in vitro experimental validation for the manuscript; N.Q., X.K., S.N., K.W., L.Z. and Y.W. discussed the results and commented on the manuscript. J.S. assisted in the completion of the revised manuscript.

## Competing interests

The authors declare no competing interests.
