## [Peer Review File · Nature Communications]

Deep Representation Learning of Chemical-induced Transcriptional Profile for Phenotype-Based Drug DiscoveryReviewer #1 (Remarks to the Author):

The manuscript presents a deep learning model TranSiGen to predict chemical-induced transcriptomics profile. Predicting chemical-induced cell type-specific transcriptomics changes is an emerging approach to virtual phenotype compound screening. Therefore, TraSiGen has the potential to serve as a valuable tool in the field of drug discovery. The primary contribution of TranSiGen involves the utilization of Variational Autoencoder (VAE) for the purpose of denoising L1000 gene expression profiles. TranSiGen outperforms several deep learning models as evidenced by benchmark studies. The manuscript is well written; nonetheless, additional proof is required to substantiate the originality and bolster the claims made within the manuscript.

1. To demonstrate the scope of TranSiGen for predicting cell type-specific drug responses, it will be interesting to see how TranSiGen performs when testing cell lines are significantly different from those in the training data, and using basal gene expression data from different resources, e.g., PANACEA and CTRP, as inputs.

2. The manuscript misses several recent published models that are quite similar to TranSiGen. For example, MultiDCP uses a Transformer based autoencoder to represent gene expressions for chemical-induced transcriptomics profile prediction [Wu et al. (2022) PLoS Comp Biol. 18(8): e1010367], which also significantly outperforms DeepCE. FAME uses contrastive learning for L1000 gene expression denoising [Pham et al. (2022) Proc SIAM Int Conf Data Min. 2022:2022:720-728. doi: 10.1137/1.9781611977172.81]. Authors should have a more thorough review in the Introduction and compare TranSiGen with these methods.

3. When evaluating the performance of TranSiGen for virtual compound screening against HTR2A, it is not clear what compound library is used. Authors should test commonly used benchmarks such as DUD-E as well as compare TranSiGen with other target-based virtual screening methods such as KarmaDock [Zhang et al. (2023) Nature Comp Sci. 3:789-804] and SOTA supervised deep learning models trained using actives/inactives of HTR2A.

Reviewer #2 (Remarks to the Author):

This manuscript entitled "TranSiGen: Deep representation learning of chemical-induced transcriptional profile" by Tong X. et al., indicated new virtual screening of chemical-induced transcriptional changes with TranSiGen. I think the idea of this article is really interesting, and the authors' fascinating observations on this timely topic may be of interest to the readers. However, some comments, as well as some crucial evidence that should be included to support the author's argumentation, needed to be addressed to improve the quality of the manuscript, its adequacy, and its readability prior to its publication in the present form.

Title: This is the most important section of the manuscript. Please present a concise and self-explanatory title stating the most important message of this article.

Abstract: I suggest the authors present the background, a short summary, and a conclusion. The general background (one to two sentences), the specific background (two to three sentences), and the current issue covered by this article (one sentence) should all be included. I would like the author to provide background information and a problem statement. The conclusion should begin with one sentence that summarizes the main message using words like "Here we highlight." The authors should describe the potential and the advancement this study has made in the field in the first sentence of the conclusion, followed by two to three sentences that provide a broader perspective.

A graphical abstract that will visually summarize the main findings of the manuscript is highly recommended. Introduction: I would like the authors to reorganize this section introduce information on the key study constructs that should be understood by readers, and make it persuasive enough to advance the primary goal of the author's recent research and the particular goal the author has intended by this article. I would like to suggest that the authors present the introduction beginning with the overall context and concluding with the current problem addressed in this article. Those key structures ought to be set up logically and coherently I also recommend that the authors provide the rationale for presenting subsequent sections.

In Fig.3a, dataset is split into 60% of train, 20% of valid, and 20% test. It was better to analyze using other split ratio. As for Dimensionality reduction, it was better to analyze using some methods including linear approach and non-linear approach, and to compare and discuss these results.

Discussion: I would like the authors to present the discussion section by opening with an introductory paragraph and followed by the summary of the previous sections. Then, I expect the authors to develop arguments clarifying the potential of this study as an extension of the previous work, the implication of the findings of this study, how this study could facilitate future research, the ultimate goal, the challenge, the knowledge and technology necessary to achieve this goal, the statement about this field in general, and finally the importance of this line of research. It is particularly important to present the limits and merit. I strongly feel the authors should dedicate a section where they explicitly discuss why and how this novel method is important for the field.

Conclusion: I think that presenting the conclusion would benefit from a single paragraph presenting some thoughtful as well as in-depth considerations. The authors should make an effort to explain the theoretical implications. I believe that it would be necessary to discuss theoretical and methodological avenues in need of refinement as well as suggestions for a path forward in understanding the importance of this study.

In performance evaluation metrics of Method section, it was better to calculate other indexes, including MAE, MAPE, MSE, MER, RMSPE, SMAPE, MRE, SSE, AIC, multiple r^2 , adjusted r^2 , and variance to discuss these performance. In ligand-based virtual screening of Method section, it was better to calculate other indexes, including balanced accuracy, F-values, logloss, and Matthews Correlation Coefficient. It was better to describe structure of encoder more detail. A structural comparison with some existing other methods is necessary to prove the feasibility of the proposed work with calculation cost, power for data size, and versatility, etc. It was better to describe future work more detail and specifically. It was better to confirm the prediction results using in vitro and in vivo analysis.

Reviewer #3 (Remarks to the Author):

In the current manuscript authors have reported a novel Deep Learning based tool called TranSiGen to infer the chemical-induced transcriptional profile of small molecules with a potential for its application in the ligand-based screening, drug response prediction and phenotypic screening.

The methodology used is sound and the results support the outcome of this research. However, some of the issues need to be resolved as follows.

1. Introduction: The biological activity of a compound depends on its cellular permeability and it should be discussed in the introduction. Likewise, few sentences on the structural representation of small molecules (ECFP) can be added to the introduction part.

2. Results: Since ECFP is a 2D descriptor representing a molecule, however, the 3D structure can contribute more towards learning how a molecule is interacting with the receptor or its target.

What was the rationale behind using ECFP and can use of a 3D descriptor or ligand structure representation make the model more predictive?

3. Methods: Overall the used methods are sound. Usage of ECFP as molecular representation can be justified.

Overall the work described in this manuscript is sound and useful to the researchers working in the field of bioinformatics and drug discovery.

REVIEWER COMMENTS

We sincerely appreciate all the reviewers for their constructive comments. We have made modifications to absorb all the suggestions and to address all the issues raised by the reviewers. In the following texts, the original comments are presented in blue with specific concerns numbered, the responses are in black, and the related changes made to our manuscript are highlighted in red.

Reviewer #1 (Remarks to the Author):

The manuscript presents a deep learning model TranSiGen to predict chemical-induced transcriptomics profile. Predicting chemical-induced cell type-specific transcriptomics changes is an emerging approach to virtual phenotype compound screening. Therefore, TraSiGen has the potential to serve as a valuable tool in the field of drug discovery. The primary contribution of TranSiGen involves the utilization of Variational Autoencoder (VAE) for the purpose of denoising L1000 gene expression profiles. TranSiGen outperforms several deep learning models as evidenced by benchmark studies. The manuscript is well written; nonetheless, additional proof is required to substantiate the originality and bolster the claims made within the manuscript.

Reply:

We sincerely thank the reviewer for thoroughly examining our manuscript and providing very helpful comments to guide our revision. We have revised the manuscript according to your suggestions and further improved the content of the article. These revisions aim to further substantiate the originality of our work and reinforce the claims presented in the manuscript.

1. To demonstrate the scope of TranSiGen for predicting cell type-specific drug responses, it will be interesting to see how TransSiGen performs when testing cell lines are significantly different from those in the training data, and using basal gene expression data from different resources, e.g., PANACEA and CTRP, as inputs.

Reply:

Thank you for your insightful comment. To elucidate the applicability of TranSiGen in predicting cell type-specific drug responses, we conducted an investigation into its performance when tested on cell lines significantly divergent from those encompassed in the training data.

In order to achieve this, we reconstructed TranSiGen utilizing a cell-blind split dataset, maintaining a train:valid:test ratio of 3:1:1 across 164 distinct cell lines. This reconstruction became imperative due to the inadequacy of response records in the test set of seven cell lines designated for evaluating TranSiGen's cell-blind performance in original manuscript.

Subsequently, a subset of the test set, composed of compounds with sufficient drug response data, was used to construct drug response models. This ensured the exclusion of cell lines present in the training set. Specifically, TranSiGen-derived representations of all compound-cell pairs were employed as input for the model training and evaluation process. **Table R1** provides details on 20 drug response testing cell lines, including their corresponding prediction performances. Notably,

although certain non-solid tumors such as K562 (chronic myelogenous leukemia), OCILY10 (diffuse large B-cell lymphoma), and NALM6 (acute lymphoblastic leukemia) exhibit distinct genetic characteristics from others [*Nature* 569, 503–508 (2019)], TranSiGen demonstrated comparable performance on these divergent cell lines.

Additionally, as illustrated in **Fig. R1**, we conducted a comparative analysis between the response model built on TranSiGen-derived representation and those constructed based on molecular structures and cell information (ECFP4+ X_1 and KPGT+ X_1). The TranSiGen-derived representation exhibited significantly superior performance, consistent with the findings presented in the original manuscript concerning cell lines included in the training set.

Table R1. The performance of drug response prediction on each cell.

Cell	Disease	Pearson
HEPG2	liver cancer	0.228±0.051
YAPC	pancreatic cancer	0.340±0.025
U2OS	osteosarcoma	0.644±0.041
T47D	breast cancer	0.895±0.008
IGR37	melanoma	0.363±0.105
SNU407	colon cancer	0.626±0.016
JHH5	liver cancer	0.223±0.027
NCIH2110	non small cell lung cancer	0.908±0.029
HEC251	endometrial cancer	0.660±0.095
K562	chronic myelogenous leukemia	0.619±0.034
BEN	non small cell lung cancer	0.636±0.126
HS578T	breast cancer	0.589±0.021
SKMEL3	melanoma	0.822±0.033
OCILY10	diffuse large B-cell lymphoma	0.482±0.059
OVK18	ovarian cancer	0.790±0.041
NALM6	acute lymphoblastic leukemia	0.672±0.036
IM95	gastric cancer	0.501±0.024
NCIH2172	non small cell lung cancer	0.966±0.028
MKN45	gastric cancer	0.668±0.097
HCT116	colon cancer	-0.078±0.148

Fig. R1 Performance of predicting drug response using various type of representations.

On the other hand, we incorporated basal gene expression profiles from different sources to assess the adaptability of our model. Specifically, we utilized basal gene expression profiles from CCLE (the origin of CTRP) as inputs for TranSiGen to infer the corresponding representation, denoted as *TranSiGen-CCLE*.

Given the distinct sequencing technologies employed in CCLE (RNA-seq) and LINCS (Affymetrix GeneChip microarrays), leading to inherent biases, we transformed CCLE profiles into a LINCS-like format. This transformation was achieved using a multi-layer perceptron (MLP) model to predict LINCS profiles from CCLE. The resultant LINCS-like CCLE basal profiles were then used to infer the representation through TranSiGen, labeled as *TranSiGen-LINCS-like-CCLE*. For clarity, *TranSiGen-LINCS* denotes the original representation derived from LINCS profiles.

Subsequently, these three types of representations were then employed in drug response prediction. As depicted in **Fig. R2**, in comparison to the model based on *TranSiGen-LINCS*, the performance of the *TranSiGen-CCLE*-based model exhibited a significant decrease. This decline may be attributed to inherent differences in the basal profiles obtained from CCLE and LINCS, limiting TranSiGen's generalization capability for predicting differential expression genes from CCLE. The MLP's ability to learn the mapping relationship from CCLE to LINCS aided in reducing the gap between data from different sources, resulting in improved performance for the *TranSiGen-LINCS-like-CCLE*-based model compared to the *TranSiGen-CCLE*-based model. However, a considerable performance gap still exists between models based on *TranSiGen-LINCS* and *TranSiGen-LINCS-like-CCLE*.

Fig. R2 Performance of predicting drug response using various type of basal profiles.

In summary, our future efforts will concentrate on addressing the heterogeneity of data from diverse sources in TranSiGen and enhancing the model's generalization performance concerning basal profiles from other platforms, such as CCLE. We have included this statement in the **Discussion** section, as shown below.

Discussion

.....

TranSiGen sets the stage for continued exploration of VAE-based models and self-supervised learning approaches in drug discovery. Our future efforts will concentrate on addressing the heterogeneity of data from diverse sources in TranSiGen and enhancing the model's generalization performance concerning basal profiles from other platforms to extend its application areas.

.....

2. The manuscript misses several recent published models that are quite similar to TranSiGen. For example, MultiDCP uses a Transformer based autoencoder to represent gene expressions for chemical-induced transcriptomics profile prediction [Wu et al. (2022) PLoS Comp Biol. 18(8): e1010367], which also significantly outperforms DeepCE. FAME uses contrastive learning for L1000 gene expression denoising [Pham et al. (2022) Proc SIAM Int Conf Data Min. 2022:2022:720-728. doi: 10.1137/1.9781611977172.81]. Authors should have a more thorough review in the Introduction and compare TranSiGen with these methods.

Reply:

Thanks for your constructive comment. First, we have added the recommended references in the **Introduction**, including MultiDCP and FAME. The relevant descriptions are as follows.

Introduction

.....

Additionally, generative models, such as Pham et al.'s FAME, applied to these perturbational profiles enable phenotypic molecular design^{10,11,12}.

.....

Furthermore, DeepCE¹⁴ and CIGER¹⁵ utilize one-hot encoding to distinguish between cell types, learning from diverse perturbational profiles. MultiDCP extends this by incorporating cellular context to predict both context-dependent gene expression and cell viability¹⁶.

.....

Moreover, we conducted a performance comparison between TranSiGen and MultiDCP, employing the experimental setup proposed in MultiDCP, which involves the challenging strategy of leaving new cells for cross-validation [*PLoS Comput. Biol.* **18**, e1010367 (2022)]. As indicated in **Supplementary Table 6**, TranSiGen outperforms MultiDCP in the inference of perturbational

profiles on new cells.

While FAME, a deep generative model for fragment-based molecular design utilizing gene expression data, focuses on denoising existing profiles to generate 64-dimensional hidden representations, it is not designed to predict unseen perturbational profiles. Therefore, a direct performance comparison between FAME and TranSiGen is not applicable.

The descriptions of performance comparison with MultiDCP are as follows.

Results

Comparison with existing models in inferring differential expression genes

.....

For cell-blind splitting, as the number of cells in the training set increases, TranSiGen demonstrates improved performance in inferring DEGs on seven unseen cells during training (Fig. 3d and Supplementary Table 5). Subsequently, we conducted a performance comparison between TranSiGen and MultiDCP, employing the experimental setup proposed in MultiDCP, which involves the challenging strategy of leaving new cells for cross-validation¹⁶. As indicated in Supplementary Table 6, TranSiGen outperforms MultiDCP in the inference of perturbational profiles on new cells.

.....

Supplementary Table 6. Model performance for inferring DEGs in cell-blind splitting.

Model	RMSE	Pearson	Positive P@100	Negative P@100
MultiDCP	1.729±0.020	0.466±0.003	0.302±0.005	0.314±0.008
TranSiGen (KPGT)	1.041±0.033	0.487±0.023	0.332±0.019	0.383±0.021

3. When evaluating the performance of TranSiGen for virtual compound screening against HTR2A, it is not clear what compound library is used. Authors should test commonly used benchmarks such as DUD-E as well as compare TranSiGen with other target-based virtual screening methods such as KarmaDock [Zhang et al. (2023) Nature Comp Sci. 3:789-804] and SOTA supervised deep learning models trained using actives/inactives of HTR2A.

Reply:

Thank you for your constructive comments. We address them point-by-point below.

(1) Compound Library Clarification.

We apologize for any confusion regarding the compound library used for HTR2A virtual screening. The descriptions of self-constructed dataset for ligand-based virtual screening have been reorganized to avoid ambiguity in the revised manuscript. The details are as follows.

Results

Ligand-based virtual screening with TranSiGen-derived representation

..... In this section, the TranSiGen-derived representation was used to assess whether a compound is active against a specific target. Specifically, we gathered and analyzed bioactivity data for compounds in LINCS2020⁶ from Pubchem⁷ (refer to Ligand-based virtual screening in the Methods section). We identified five distinct targets, namely HTR2A, DRD2, ADRA2A, SLC6A4, and KCNH2, each having a significant number of active compounds, as illustrated in Supplementary Fig. 4. This allowed us to construct active/inactive random forest (RF) classifiers for each target. The representations of compounds were based on the DEGs inferred by TranSiGen and other baseline models.

Methods

Ligand-based virtual screening

We constructed a compound library (7,148 compounds) for virtual screening against HTR2A. This library excluded the 355 compounds used for parallel performance comparison across seven cell lines, ensuring data integrity and preventing leakage. To assess model performance in downstream tasks, we predicted perturbation profiles for these 7,148 compounds, which were then used to train the models. Target annotations for these molecules were obtained from two sources: the LINCS2020²² compound information file and PubChem⁴⁴ bioactivity data. For PubChem data, we considered compounds with IC₅₀, Ki, or K_d values below 10 μM as targets. This process identified 41 active compounds for HTR2A. The remaining compounds in the external test set were randomly sampled at a 1:5 ratio to create a set of inactive compounds. Notably, similar screening was also conducted for four additional targets (DRD2, ADRA2A, SLC6A4, and KCNH2) using the same strategy as HTR2A. Finally, the compounds were randomly split into training and test set with a ratio of 4:1 in each cell line.

.....

(2) Benchmark Testing and Comparison.

Ligand-based virtual screening is indeed a promising application for TranSiGen-derived representations. As demonstrated with our self-constructed dataset, this representation effectively identifies new scaffold compounds (**Fig 4d and Supplementary Fig. 6**). However, we believe that directly comparing TranSiGen with methods like KPGT on benchmarks like DUD-E may not be entirely suitable. Unlike our dataset, DUD-E datasets often have highly similar active compounds and easily identifiable decoys based on structural information (**Fig. R3**). While our model can prioritize the actives against decoys better than ECFP4 for compounds with significant structural differences in such scenarios (**Fig. R4**), KPGT generally performs better (**Fig. R5 and Table R2**). TranSiGen focuses on capturing the phenotypic response of compounds within a biological environment, whereas KPGT prioritizes target prediction. This difference in focus makes direct performance comparisons challenging due to the complex interplay between compounds and cellular targets.

Fig. R3 Dimensionality reduction visualization of compound datasets for different targets based on ECFP4 using UMAP.

Fig. R4 The rank of prediction results for compounds with a chemical structure similarity $\in (0.0, 0.3]$ to the training compounds. Real labels of compounds are used to colour the actives and decoys, respectively.

Table R2. Model performance of ligand-based virtual screening on DUD-E.

	AUROC	AUPR	BACC	F1	Log_loss	MCC	Recall	Precision
ECFP4	0.985	0.917	0.951	0.694	0.045	0.718	0.921	0.604
	± 0.021	± 0.098	± 0.039	± 0.216	± 0.079	± 0.189	± 0.068	± 0.258
KPGT	0.993	0.939	0.943	0.69	0.107	0.714	0.903	0.605
	± 0.017	± 0.078	± 0.097	± 0.26	± 0.254	± 0.24	± 0.194	± 0.296
TranSiGen_LF	0.97	0.747	0.921	0.44	0.081	0.496	0.891	0.315
	± 0.027	± 0.168	± 0.043	± 0.187	± 0.061	± 0.162	± 0.069	± 0.183

Fig. R5 Model performance of ligand-based virtual screening on self-constructed dataset.

(3) Comparison with SOTA models.

Target-based methods such as KarmaDock are orthogonal our approach. While KarmaDock leverages the protein's structural data, our method uses ligand-induced transcriptomic perturbation data. Since these two methods operate under completely different assumptions, they cannot be compared in similar settings.

Besides, according to your suggestions, we compare our TranSiGen-derived HTR2A RF classifier with SOTA supervised models trained using actives/inactives of HTR2A, including Xgboost models based on molecular structural representations like ECFP4, KPGT, and 3D descriptors, as well as the newly interpretable GNN-based model Substructure-Mask Explanation (SME) [*Nat. Commun.* **14**, 2585 (2023)], Graph Isomorphism Network (GIN) [*arXiv preprint arXiv.1710.10903* (2017)], and Graph Attention Network (GAT) [*stat.* **1050(20)**, 10-48550 (2017)]. The performance of these models in screening active compounds at various maximum similarity thresholds is shown in **Fig. R6** and **Table R3**.

The overall comparison results for various models align with the trends discussed in the original manuscript, demonstrating the advantages of TranSiGen-based models in screening novel scaffold compounds. Specifically, when analyzing compounds dissimilar to the training set (chemical structure similarity $\in (0.0, 0.3]$), the TranSiGen-based model exhibits superior predictive capability, particularly compared to machine learning models based on predefined representations. And for compounds similar to the training set (chemical structure similarity $\in (0.3, 1.0]$), all models show comparable performance.

Fig. R6 Model performance of ligand-based virtual screening on target HTR2A within different thresholds of max similarity of test molecules relative to train data.

Table R3. Model performance of ligand-based virtual screening on target HTR2A within different thresholds of max similarity of test molecules relative to train data.

Structure similarity	Model	AUROC	AUPR	BACC	F1	Log_loss	MCC
(0.0, 0.3]	RF (TranSiGen_LF)	0.977±0.010	0.856±0.056	0.846±0.010	0.700±0.046	0.261±0.003	0.657±0.053
	Xgboost (ECFP4)	0.961±0.01	0.608±0.131	0.846±0.01	0.7±0.046	0.373±0.003	0.657±0.053
	Xgboost (Descriptors_3D)	0.846±0.026	0.682±0.054	0.732±0.0	0.571±0.0	0.287±0.005	0.527±0.0
	Xgboost (KPGT)	0.88±0.046	0.502±0.125	0.768±0.049	0.478±0.097	0.268±0.033	0.409±0.11
	SME	0.846±0.081	0.467±0.072	0.704±0.06	0.391±0.074	0.619±0.142	0.302±0.097
	GAT	0.961±0.017	0.742±0.08	0.832±0.03	0.653±0.106	0.452±0.213	0.606±0.118
	GIN	0.946±0.018	0.703±0.088	0.821±0.013	0.602±0.043	0.639±0.143	0.548±0.047
(0.3, 1.0]	RF (TranSiGen_LF)	0.955±0.024	0.921±0.043	0.845±0.024	0.801±0.032	0.543±0.007	0.700±0.062
	Xgboost (ECFP4)	0.961±0.017	0.927±0.031	0.862±0.02	0.821±0.029	0.495±0.007	0.72±0.049
	Xgboost (Descriptors_3D)	0.958±0.007	0.914±0.017	0.871±0.0	0.833±0.0	0.358±0.007	0.742±0.0
	Xgboost (KPGT)	0.991±0.008	0.985±0.014	0.889±0.025	0.864±0.041	0.283±0.04	0.795±0.072
	SME	0.945±0.017	0.902±0.036	0.829±0.042	0.778±0.053	0.647±0.075	0.665±0.072
	GAT	0.927±0.02	0.859±0.042	0.826±0.0	0.769±0.0	0.627±0.161	0.633±0.0
	GIN	0.97±0.0	0.946±0.005	0.864±0.017	0.827±0.015	0.395±0.05	0.744±0.004

Reviewer #2 (Remarks to the Author):

This manuscript entitled "TranSiGen: Deep representation learning of chemical-induced transcriptional profile" by Tong X. et al., indicated new virtual screening of chemical-induced transcriptional changes with TranSiGen. I think the idea of this article is really interesting, and the authors' fascinating observations on this timely topic may be of interest to the readers. However, some comments, as well as some crucial evidence that should be included to support the author's argumentation, needed to be addressed to improve the quality of the manuscript, its adequacy, and its readability prior to its publication in the present form.

Reply:

Thank you very much for your recognition and evaluation of our work. We have revised the manuscript according to your suggestions and further improved the content of the article to ensure its quality and reliability.

1. Title: This is the most important section of the manuscript. Please present a concise and self-explanatory title stating the most important message of this article.

Reply:

Thank you for your valuable suggestion. We've updated the title to "Deep Representation Learning of Chemical-induced Transcriptional Profile for Phenotype-Based Drug Discovery" to better emphasize the potential of our research for advancing this important area of drug discovery.

2. Abstract: I suggest the authors present the background, a short summary, and a conclusion. The general background (one to two sentences), the specific background (two to three sentences), and the current issue covered by this article (one sentence) should all be included. I would like the author to provide background information and a problem statement. The conclusion should begin with one sentence that summarizes the main message using words like "Here we highlight." The authors should describe the potential and the advancement this study has made in the field in the first sentence of the conclusion, followed by two to three sentences that provide a broader perspective.

Reply:

Thank you for your valuable suggestion. In the revised manuscript, we have refined the abstract in accordance with the suggested logic to ensure that the background, summary of results, and a conclusion are all clearly included. The details are as follows.

Abstract

Artificial intelligence transforms drug discovery, with phenotype-based approaches emerging as a promising alternative to target-based methods, overcoming limitations like lack of well-defined targets. While chemical-induced transcriptional profiles offer a comprehensive view of drug mechanisms, inherent noise often obscures the true signal, hindering their potential for meaningful insights. Here, we highlight the development of TranSiGen, a deep generative model employing self-supervised representation learning. TranSiGen analyzes basal cell gene expression and molecular structures to reconstruct chemical-induced transcriptional profiles with high accuracy. By

capturing both cellular and compound information, TranSiGen-driven representations demonstrate efficacy in diverse downstream tasks like ligand-based virtual screening, drug response prediction, and phenotype-based drug repurposing. Notably, in vitro validation of TranSiGen's application in pancreatic cancer drug discovery highlights its potential for identifying effective compounds. We envisage that integrating TranSiGen into the drug discovery and mechanism research holds significant promise for advancing biomedicine.

3. A graphical abstract that will visually summarize the main findings of the manuscript is highly recommended.

Reply:

Thank you for your valuable suggestion. We acknowledge the importance of a clear and concise visual representation of our research. Addressing your feedback, we have included a graphical abstract (**Fig. R7**) within the manuscript. It visually summarizes the key steps involved in using TranSiGen to predict perturbational profiles, along with the potential applications in downstream tasks.

Fig R7. The graphical abstract of TranSiGen.

4. Introduction: I would like the authors to reorganize this section introduce information on the key study constructs that should be understood by readers, and make it persuasive enough to advance the primary goal of the author's recent research and the particular goal the author has intended by this article. I would like to suggest that the authors present the introduction beginning with the overall context and concluding with the current problem addressed in this article. Those key structures ought to be set up logically and coherently I also recommend that the authors provide the rationale for presenting subsequent sections.

Reply:

Thank you for your valuable feedback. We agree that a well-structured Introduction is key to setting the context and engaging readers. Accordingly, we have thoroughly reorganized the **Introduction** to begin with the broader importance of transcriptomics data analysis in drug discovery. It then logically transitions to the specific challenges faced by existing transcriptional profile predictors, namely the difficulty in distinguishing true perturbation signals from inherent noise. This revised section will logically and coherently introduce key study constructs essential for reader understanding. Additionally, it will provide a clear rationale for the content presented in subsequent sections.

Introduction

The field of drug discovery is experiencing a paradigm shift driven by artificial intelligence (AI). While target-based approaches have long dominated the field, their limitations—including a lack of well-defined targets, off-target effects, and unsatisfactory therapeutic responses—have driven the rise of phenotype-based methods. These approaches focus on the comprehensive cellular response to drug candidates, offering a more holistic understanding of disease mechanisms and potentially revealing novel drug targets and therapeutic avenues.

Transcriptomics data analysis plays a crucial role in drug discovery and understanding disease mechanisms. By capturing the global gene expression landscape across diverse biological contexts, it offers rich insights into cellular and organismal states. High-throughput RNA sequencing (RNA-seq) technologies have facilitated the generation of large-scale perturbational gene expression profiles, exemplified by databases like Connectivity Map (CMap)¹, the Library of Integrated Network-based Cell-Signature (LINCS)², PANACEA³, ARCHS4⁴ and ChemPert⁵. These large-scale perturbational gene expression profiles provide invaluable insights into how cells respond to various disruptions. The exploration of these profiles plays a central role in drug discovery, helping to elucidate mechanisms of action (MOA)^{1,6,7}. For example, the CMap project proposed a pattern-matching strategy to identify compounds with shared MOA¹. Furthermore, machine learning models like our proposed SSGCN can analyze relationships between chemical-induced and gene knockdown-induced profiles, offering a powerful method for identifying potential drug targets^{8,9}. Additionally, generative models, such as Pham et al.'s FAME, applied to these perturbational profiles enable phenotypic molecular design^{10,11,12}.

Despite the immense value of perturbational gene expression profiles, the combinatorial complexity of drug-like molecules and cell lines limits exhaustive exploration through high-throughput experiments. This challenge has spurred the development of deep learning models capable of predicting transcriptional profiles for novel chemicals using publicly available data. DLEPS is a deep neural network designed to predict gene expression responses to new chemicals without cell-type specificity¹³. Furthermore, DeepCE¹⁴ and CIGER¹⁵ utilize one-hot encoding to distinguish between cell types, learning from diverse perturbational profiles. MultiDCP extends this by incorporating cellular context to predict both context-dependent gene expression and cell viability¹⁶.

However, supervised learning models that directly fit gene expression values may struggle to distinguish true perturbation signals from confounding factors and the inherent noise within expression profiles. Recent studies highlight the power of variational autoencoders (VAEs) in handling high-dimensional, noisy transcriptomics data^{17,18}. To address the limitations of data and generate novel perturbational profiles, we propose **Transcriptional Signatures Generator**

(TranSiGen), a VAE-based framework leverages self-supervised representation learning to denoise and reconstruct transcriptional profiles, enabling the inference of new perturbational profiles. TranSiGen simultaneously learns three key distributions: the basal profiles without perturbation, the induced perturbational profiles, and the mapping relationship between them. This self-supervised approach effectively mitigates noise in the data and uncovers the underlying perturbation signals. TranSiGen offers several key benefits. (1) Improved inference of transcriptional profiles: TranSiGen's superior performance in inferring basal profiles, chemical-perturbational profiles, and the corresponding differential expression genes (DEGs) was demonstrated by comparisons with baseline models. (2) Unified representation for cellular and compound features: TranSiGen's generated perturbational profiles effectively capture both cellular and compound features, as evidenced by visualization analysis differentiating cell lines and drugs' MOA. (3) Versatile applications in downstream tasks: TranSiGen-derived representations have proven effective in various tasks including ligand-based virtual screening, drug response prediction, and phenotype-based drug repurposing. Its application in screening drugs against pancreatic cancer, with subsequent in vitro validation and high hit rates, demonstrates the power of TranSiGen's phenotype-based approach for identifying potent compounds. Importantly, TranSiGen's integration into phenotype-based drug discovery pipelines has the potential to significantly improve efficiency and reduce costs.

5. In Fig.3a, dataset is splitted into 60% of train, 20% of valid, and 20% test. It was better to analyze using other split ratio.

Reply:

Thank you for your insightful comments. In the original manuscript, the dataset was divided into 60% for training, 20% for validation, and 20% for testing. This division was reiterated five times using distinct random seeds, serving to mitigate randomness and ensure a robust evaluation of the model's performance.

In response to your suggestion, we adopted a five-fold cross-validation approach, designating each fold as a test set, while utilizing the remaining data for training and evaluation. This modification allowed for a comprehensive comparison of the models' performance across five distinct test sets, as detailed in **Supplementary Table 2**.

The overall comparison results for various models align with the trends mentioned in the original manuscript. Specifically, in scenario 2-1, TranSiGen surpasses the other three models (DLEPS, DeepCE, and CIGER) in terms of average PCC evaluated on seven cell lines. In scenario 2-2, TranSiGen attains state-of-the-art results on the full dataset, which encompasses a broader range of cell lines and molecules. In addition, the optimal setting for TranSiGen's performance in inferring DEGs remains unchanged, with molecular representations utilizing KPGT and parameter initialization based on perturbational profiles from gene knockdown.

The descriptions of **Supplementary Table 2** have been included in the revised manuscript as follow.

Results

Comparison with existing models in inferring differential expression genes

.....

Scenario 2 results are shown in Fig. 3b and 3c, detailed in Supplementary Tables 1 for time consumption and computational resources, and Supplementary Tables 2 and 3 for metric scores. TranSiGen performs better than the other three models (DLEPS, DeepCE and CIGER) in chemical-blind (scenario 2-1, Fig. 3b), in terms of the average PCC (represented by the black dots) evaluated on seven cell lines. Additionally, TranSiGen, along with DeepCE and CIGER that are also capable of inferring cell-specific profiles, shows similar trends in the prediction performance of DEGs on different cell lines. TranSiGen also outperforms the above three models in terms of Positive Precision@100 and Negative Precision@100 when evaluating metrics focusing on the most significantly up- and down-regulated expressed genes, **while exhibits comparable computational cost with other models.**

.....

Supplementary Table 2. Model performance for inferring DEGs in chemical-blind setting at 5-fold cross validation.

Data	Model	RMSE	Pearson	Positive P@100	Negative P@100
seven cell lines (scenario 2-1)	DLEPS	1.495±0.040	0.427±0.019	0.246±0.013	0.309±0.012
	DeepCE	1.773±0.034	0.412±0.005	0.228±0.005	0.271±0.007
	CIGER	4.465±1.061	0.416±0.012	0.234±0.015	0.285±0.008
	TranSiGen (ECFP4)	0.690±0.022	0.533±0.016	0.379±0.012	0.394±0.012
	TranSiGen (KPGT)	0.679±0.025	0.546±0.013	0.389±0.009	0.400±0.012
full data (scenario 2-2)	TranSiGen (ECFP4; init_random)	0.527±0.003	0.607±0.002	0.434±0.004	0.442±0.001
	TranSiGen (ECFP4)	0.522±0.004	0.613±0.002	0.439±0.002	0.445±0.002
	TranSiGen (KPGT; init_random)	0.523±0.005	0.611±0.003	0.435±0.003	0.444±0.003
	TranSiGen (KPGT)	0.518±0.003	0.617±0.002	0.443±0.002	0.450±0.001

6. As for Dimensionality reduction, it was better to analyze using some methods including linear approach and non-linear approach, and to compare and discuss these results.

Reply:

Thank you for your constructive suggestion. In response to your suggestion, we employed various dimensionality reduction techniques to analyze the data, including principal component analysis (PCA) for linear approaches and t-distributed stochastic neighbor embedding (t-SNE) and uniform

manifold approximation and projection (UMAP) for non-linear approaches (**Fig. R8**).

While visual analysis revealed limitations in distinguishing active and inactive compounds using PCA, TranSiGen-derived representations exhibited clear segregation based on activity in the non-linearity-based t-SNE and UMAP visualizations. This distinction arises from the manifold-learning capabilities of techniques like t-SNE and UMAP, which excel in capturing underlying data structures compared to PCA projection. Notably, UMAP's visualization demonstrates even clearer clustering of active compounds compared to t-SNE, suggesting its potential for superior interpretability. In contrast, other perturbational representations (DLEPs, DeepCE, and CIGER) displayed overlapping distributions for active and inactive compounds across various dimensionality reduction techniques.

Fig. R8 Dimensionality reduction visualization of HTR2A active and inactive compounds based on various inferred perturbational representations.

7. Discussion: I would like the authors to present the discussion section by opening with an introductory paragraph and followed by the summary of the previous sections. Then, I expect the authors to develop arguments clarifying the potential of this study as an extension of the previous

work, the implication of the findings of this study, how this study could facilitate future research, the ultimate goal, the challenge, the knowledge and technology necessary to achieve this goal, the statement about this field in general, and finally the importance of this line of research. It is particularly important to present the limits and merit. I strongly feel the authors should dedicate a section where they explicitly discuss why and how this novel method is important for the field.

Conclusion: I think that presenting the conclusion would benefit from a single paragraph presenting some thoughtful as well as in-depth considerations. The authors should make an effort to explain the theoretical implications. I believe that it would be necessary to discuss theoretical and methodological avenues in need of refinement as well as suggestions for a path forward in understanding the importance of this study.

Reply:

We have revised the Discussion and Conclusion sections based on your valuable feedback. Following the conventional article structure of Nature Communications, we combined these two sections into a single “**Discussion**” section.

Discussion

The field of drug discovery is undergoing a significant transformation with the integration of AI. While target-based approaches have been the dominant strategy, they are often hampered by limitations: (1) Many diseases lack clear and accessible protein targets, making it difficult to design effective drugs. (2) Targeting specific proteins can inadvertently interact with other molecules in the cell, leading to off-target effects and unexpected side effects. (3) Even when a target is identified, the resulting drug may struggle to reach its target within the cell due to poor cell permeability, thereby hindering the achievement of the desired therapeutic effect^{38,39}. These challenges have spurred the emergence of phenotype-based approaches, which directly analyze overall cellular response to drugs, offering a more holistic understanding of disease mechanisms, and the potential for discoveries of novel drug mechanisms and therapeutic opportunities^{13,40}.

This study introduced TranSiGen, a VAE-based framework designed to address the limitations of supervised learning models for perturbational gene expression data. TranSiGen’s novel self-supervised representation learning strategy effectively denoises transcriptional profiles. Extensive evaluations show that TranSiGen outperforms existing models in inferring basal profiles, chemical-induced perturbational profiles, and corresponding DEGs. This capability unlocks new avenues for expanding and enhancing existing drug discovery datasets. TranSiGen’s core strength lies in its ability to overcome the noise and confounding factors inherent in gene expression profiles, offering a standardized way to characterize phenotypic information related to both cellular context and compound effects. This standardization facilitates integration and efficiency improvement across various downstream tasks, including ligand-based virtual screening, drug response prediction, and phenotype-based drug repurposing. Notably, its use in phenotype-based drug repurposing for pancreatic cancer, with subsequent in vitro validation, showcases its promise for real-world drug discovery scenarios.

TranSiGen sets the stage for continued exploration of VAE-based models and self-supervised learning approaches in drug discovery. Our future efforts will concentrate on addressing the heterogeneity of data from diverse sources in TranSiGen and enhancing the model’s generalization

performance concerning basal profiles from other platforms to extend its application areas. Additionally, we plan to enhance the model's precision and interpretability by incorporating prior biological knowledge, such as pathways and gene ontologies. Beyond its current application in drug discovery, we are eager to investigate TranSiGen's potential utility in precision medicine and disease modeling, recognizing the substantial promise in these areas. The ultimate goal in this field is to create a truly comprehensive framework for efficiently utilizing high-dimensional gene expression data. This will accelerate drug discovery and unravel the complexities of disease mechanisms. TranSiGen, with its unique strengths and extensibility, marks a valuable step toward realizing this goal.

8. In performance evaluation metrics of Method section, it was better to calculate other indexes, including MAE, MAPE, MSE, MER, RMSPE, SMAPE, MRE, SSE, AIC, multiple r^2 , adjusted r^2 , and variance to discuss these performance.

Reply:

Thank you for your valuable feedback. In response to your suggestion, we expanded the range of metrics considered beyond RMSE, Pearson correlation, Positive P@100, and Negative P@100, as presented in the original paper. Additional metrics, including **SSE**, **MSE**, **MAE**, and **multiple r^2** , were calculated, and the corresponding results are detailed in **Supplementary Tables 3, 4, and 5**.

Overall, the comparative analysis of various models aligns with the trends outlined in the original manuscript. In the chemical-blind setting, TranSiGen demonstrates superior performance across all metrics compared to the other three models (DLEPS, DeepCE, and CIGER). Specifically, TranSiGen achieves state-of-the-art results on the comprehensive dataset, encompassing a broader array of cell lines and molecules (refer to **Supplementary Table 3**). A similar trend is observed in the random splitting scenario (refer to **Supplementary Table 4**). In instances of cell-blind splitting, the enhanced performance of TranSiGen in inferring DEGs across all metrics becomes more pronounced as the number of cells in the training set increases (refer to **Supplementary Table 5**).

We excluded the metrics MAPE, RMSPE, SMAPE, MRE, and MER from consideration for this study due to the presence of zero values in both the true and predicted perturbational profiles. Furthermore, we did not include AIC and adjusted r^2 , typically applied in linear regression models. The inherent complexity of deep learning models poses challenges in adequately characterizing them through these metrics. Nevertheless, the upgraded metrics are sufficient to fulfill the requirement to assess the models' performance from both overall and specific perspectives. The metrics SSE, MSE, MAE, and RMSE quantify the disparity between predicted and true values, while Pearson coefficient and multiple r^2 gauge their linear relationship. These metrics assess the prediction performance concerning the overall landmark genes. On the other hand, Positive P@100 and Negative P@100 focus on the most significantly up-regulated and down-regulated genes.

Supplementary Table 3. Model performance for inferring DEGs in chemical-blind setting.

Data	Model	RMSE	Pearson	Positive P@100	Negative P@100	SSE	MSE	MAE	multiple r ²
seven cell lines (scenario 2-1)	DLEPS	1.551±0.005	0.418±0.004	0.244±0.002	0.316±0.002	2453.931±10.836	2.509±0.011	1.256±0.001	0.155±0.004
	DeepCE	1.773±0.034	0.43±0.02	0.238±0.003	0.29±0.01	3202.744±142.18	3.275±0.145	1.428±0.026	0.122±0.046
	CIGER	2.551±0.282	0.436±0.002	0.258±0.017	0.287±0.001	6647.359±1444.233	6.797±1.477	2.056±0.244	-1.016±0.464
full data (scenario 2-2)	TranSiGen (ECP4)	0.661±0.003	0.517±0.001	0.363±0.002	0.387±0.003	470.711±4.38	0.481±0.004	0.489±0.004	-0.022±0.029
	TranSiGen (KPGT)	0.641±0.002	0.54±0.002	0.381±0.002	0.397±0.001	446.824±2.102	0.457±0.002	0.474±0.001	0.048±0.016
	TranSiGen (ECP4; init_random)	0.527±0.004	0.609±0.001	0.433±0.002	0.442±0.0	314.004±6.022	0.321±0.006	0.366±0.003	0.316±0.008
full data (scenario 2-2)	TranSiGen (ECP4)	0.522±0.005	0.615±0.002	0.441±0.0	0.448±0.004	309.535±6.284	0.316±0.006	0.362±0.004	0.328±0.004
	TranSiGen (KPGT; init_random)	0.524±0.004	0.613±0.001	0.437±0.002	0.445±0.002	310.245±5.429	0.317±0.006	0.364±0.002	0.322±0.005
	TranSiGen (KPGT)	0.52±0.003	0.619±0.0	0.443±0.003	0.452±0.002	306.549±5.027	0.313±0.005	0.36±0.002	0.334±0.004

Supplementary Table 4. Model performance for inferring DEGs in random splitting.

Data	Model	RMSE	Pearson	Positive P@100	Negative P@100	SSE	MSE	MAE	multiple r ²
seven cell lines	DeepCE	1.705±0.014	0.496±0.009	0.267±0.004	0.319±0.006	2979.628±54.194	3.047±0.055	1.384±0.012	0.217±0.015
	CIGER	3.421±1.248	0.512±0.006	0.3±0.008	0.326±0.007	14956.058±9500.647	15.292±9.714	2.79±1.051	-3.16±2.717
seven cell lines	TranSiGen (ECFP4)	0.615±0.008	0.626±0.005	0.449±0.005	0.465±0.005	425.375±13.556	0.435±0.014	0.449±0.006	0.242±0.005
	TranSiGen (KPGT)	0.61±0.006	0.631±0.006	0.453±0.008	0.467±0.005	418.155±10.22	0.428±0.01	0.445±0.004	0.254±0.013
full data	TranSiGen (ECFP4; init_random)	0.502±0.001	0.639±0.001	0.46±0.001	0.469±0.001	278.842±2.307	0.285±0.002	0.346±0.001	0.36±0.004
	TranSiGen (ECFP4)	0.501±0.002	0.64±0.001	0.459±0.001	0.471±0.001	277.304±2.489	0.284±0.003	0.345±0.001	0.366±0.004

Supplementary Table 5. Model performance for inferring DEGs in cell-blind splitting.

Model	RMSE	Pearson	Positive P@100	Negative P@100	SSE	MSE	MAE	multiple r ²
TranSiGen (KPGT; 10 cells)	1.198±0.004	0.26±0.001	0.214±0.001	0.2±0.002	1455.064±9.649	1.488±0.01	0.884±0.003	-3.685±0.019
TranSiGen (KPGT; 50 cells)	1.073±0.008	0.297±0.003	0.241±0.002	0.223±0.004	1146.384±14.134	1.172±0.014	0.791±0.006	-2.761±0.006
TranSiGen (KPGT; 150 cells)	0.96±0.012	0.324±0.003	0.256±0.002	0.232±0.004	917.258±23.053	0.938±0.024	0.701±0.013	-1.878±0.081

9. In ligand-based virtual screening of Method section, it was better to calculate other indexes, including balanced accuracy, F-values, logloss, and Matthews Correlation Coefficient.

Reply:

Thank you for your valuable feedback. In response to your comment, we expanded our assessment by including balanced accuracy (**BACC**), F-values (**F1**), **Log_loss**, and matthews correlation coefficient (**MCC**) in the ligand-based virtual screening task. The corresponding results can be found in **Supplementary Tables 7 and 8**. Illustratively, focusing on the ligand-based virtual screening of HTR2A, the model utilizing TranSiGen-derived representations surpassed models employing alternative perturbational representations across all metrics (refer to **Supplementary Table 7**). Furthermore, in comparison to models utilizing molecular structure representations (specifically, molecular fingerprint ECFP4, and pre-trained representation KPGT), the TranSiGen-based model demonstrated superior predictive capability for compounds dissimilar to the training set, as indicated by chemical structure similarity $\in (0.0, 0.3]$ in **Supplementary Table 8**.

Supplementary Table 7. Model performance of ligand-based virtual screening on target HTR2A using different perturbational representations.

Model	AUROC	AUPR	BACC	F1	Log_loss	MCC
DLEPS	0.566±0.000	0.265±0.000	0.583±0.000	0.329±0.000	0.688±0.000	0.164±0.000
DeepCE	0.527±0.041	0.225±0.030	0.557±0.041	0.315±0.074	0.635±0.097	0.102±0.066
CIGER	0.614±0.086	0.397±0.105	0.599±0.057	0.366±0.090	0.518±0.076	0.214±0.124
TranSiGen	0.889±0.041	0.731±0.092	0.784±0.047	0.620±0.076	0.369±0.022	0.523±0.103

Supplementary Table 8. Model performance of ligand-based virtual screening on target HTR2A within different thresholds of max similarity of test molecules relative to train data.

Structure similarity	Model	AUROC	AUPR	BACC	F1	Log_loss	MCC
(0.0, 0.3]	ECFP4	0.910±0.048	0.568±0.105	0.750±0.081	0.534±0.127	0.271±0.015	0.467±0.155
	KPGT	0.856±0.049	0.507±0.160	0.739±0.041	0.445±0.077	0.314±0.013	0.372±0.087
	TranSiGen (late fusion)	0.977±0.010	0.856±0.056	0.846±0.010	0.700±0.046	0.261±0.003	0.657±0.053
(0.3, 1.0]	ECFP4	0.938±0.014	0.852±0.063	0.847±0.054	0.800±0.075	0.416±0.027	0.719±0.052
	KPGT	0.988±0.013	0.978±0.023	0.889±0.025	0.864±0.041	0.381±0.013	0.795±0.072
	TranSiGen (late fusion)	0.955±0.024	0.921±0.043	0.845±0.024	0.801±0.032	0.543±0.007	0.700±0.062

The bold font signifies the optimal performance at the similarity thresholds (0.0, 0.3] and (0.3, 1.0].

10. It was better to describe structure of encoder more detail.

Reply:

Thank you for your constructive suggestion. In response, we have revised the manuscript to provide a more detailed explanation of the encoder in the VAE model. The corresponding description in the revised manuscript is as follows.

Methods

TranSiGen architecture

The VAE⁴³ is a deep generative model consisting of an encoder and a decoder. **The encoder extracts significant information from the input, compressing it into a latent representation. Meanwhile, the decoder reconstructs a near-identical output from this latent vector.** Consequently, VAE is capable of learning an efficient and meaningful latent space from high-dimensional data by compressing and reconstructing the original input. Unlike the standard autoencoder, which maps the input to a point in the latent space and trains by minimizing the reconstruction error, VAE encodes the input to a distribution. This requires the addition of a Kullback-Leibler (KL) divergence term to the reconstruction loss, which constrains the latent vectors to match a Gaussian distribution.

The architecture of TranSiGen consists of two VAEs: one for encoding the basal profiles X_1 and the other for encoding the perturbation profiles X_2 . **Each VAE comprises an encoder with two hidden layers ([1200, 100] dimensions) and a corresponding two-layer decoder ([100, 800] dimensions).** TranSiGen minimizes the loss of learning the representations of X_1 and X_2 . Additionally, a linear function is used to map from the latent representation Z_1 of X_1 and the hidden representation Z_{mol} of the compound representation C_{mol} to the perturbed latent representation $Z_2 F Z_1$ of X_2' , mimicking the chemical-induced transcription changes. The layer and dimension details of TranSiGen are shown in Supplementary Fig. 1.

11. A structural comparison with some existing other methods is necessary to prove the feasibility of the proposed work with calculation cost, power for data size, and versatility, etc.

Reply:

Thank you for your valuable insights. In response to your comment, we have calculated the number of parameters and inference time for each model investigated. Recognizing their importance in assessing computational requirements and efficiency, we have included these results in **Supplementary Table 1**.

TranSiGen exhibits a favorable balance between performance and computational cost. While it has a slightly higher parameter count than the smallest model, it achieves the fastest inference time among the compared methods. These results, in conjunction with TranSiGen's superior performance highlighted in the manuscript, demonstrate its efficiency and overall strength as a tool for computational drug discovery.

The description of **Supplementary Table 1** in the revised manuscript is as follow.

Results

Comparison with existing models in inferring differential expression genes

.....

Scenario 2 results are shown in Fig. 3b and 3c, detailed in Supplementary Tables 1 for time consumption and computational resources, and Supplementary Tables 2 and 3 for metric scores. TranSiGen performs better than the other three models (**DLEPS, DeepCE and CIGER**) in chemical-blind (scenario 2-1, Fig. 3b), in terms of the average PCC (represented by the black dots) evaluated

on seven cell lines. Additionally, TranSiGen, along with DeepCE and CIGER that are also capable of inferring cell-specific profiles, shows similar trends in the prediction performance of DEGs on different cell lines. TranSiGen also outperforms the above three models in terms of Positive Precision@100 and Negative Precision@100 when evaluating metrics focusing on the most significantly up- and down-regulated expressed genes, **while exhibits comparable computational cost with other models.**

.....

Supplementary Table 1. The number of parameters for each model.

Model	#Params (MB)	Time (seconds)
DLEPS	20.441	0.019±0.001
DeepCE	2.605	0.021±0.003
CIGER	17.627	0.017±0.004
TranSiGen	15.915	0.009±0.001

12. It was better to describe future work more detailand specifically.

Reply:

Thank you for your valuable suggestion. We have expanded the **Discussion** section to address future research directions. This includes exploring strategies to handle data heterogeneity, incorporating relevant biological knowledge, and investigating TranSiGen’s potential applications in precision medicine and disease modeling.

Discussion

.....

TranSiGen sets the stage for continued exploration of VAE-based models and self-supervised learning approaches in drug discovery. Our future efforts will concentrate on addressing the heterogeneity of data from diverse sources in TranSiGen and enhancing the model’s generalization performance concerning basal profiles from other platforms to extend its application areas. Additionally, we plan to enhance the model’s precision and interpretability by incorporating prior biological knowledge, such as pathways and gene ontologies. Beyond its current application in drug discovery, we are eager to investigate TranSiGen’s potential utility in precision medicine and disease modeling, recognizing the substantial promise in these areas. The ultimate goal in this field is to create a truly comprehensive framework for efficiently utilizing high-dimensional gene expression data. This will accelerate drug discovery and unravel the complexities of disease mechanisms. TranSiGen, with its unique strengths and extensibility, marks a valuable step toward realizing this goal.

13. It was better to confirm the prediction results using in vitro and in vivo analysis.

Reply:

Thank you for your constructive comments. In addition to the original literature validation, we

conducted in vitro experimental analyses to validate the predicted outcomes in phenotype-based drug repurposing for pancreatic cancer treatment. The results demonstrated high hit rates, with TranSiGen_DISEASE and TranSiGen_DRUG achieving rates of 38% and 80%, respectively. These findings emphasize the effectiveness of phenotype-based strategies utilizing TranSiGen-derived representations in identifying potent candidate compounds. The corresponding description in the revised manuscript is as follows.

Results

Phenotype-based drug repurposing for the treatment of pancreatic cancer

.....

Moreover, we conducted a phenotype-based screening using the compound library available in our laboratory for pancreatic cancer and subsequently experimentally validated the top candidates in vitro. Specifically, we employed TranSiGen to predict the DEGs of 31,465 compounds on the YAPC pancreatic cancer cell line. These compounds were ranked using two distinct phenotype-based strategies, TranSiGen_DISEASE and TranSiGen_DRUG. Subsequently, we selected the top 50 compounds from each strategy to experimentally assess their activity on YAPC cells. Notably, positive molecules, including chemotherapy drugs (fluorouracil, gemcitabine, mitomycin C, paclitaxel and irinotecan), and targeted drugs (sunitinib, erlotinib and everolimus), were included as reference compounds (refer to the “Experimental setting of validating screened compounds for pancreatic cancer”).

Figures 6e and 6f display the top 50 compounds identified through screening with TranSiGen_DISEASE and TranSiGen_DRUG, along with their respective cell proliferation inhibition activities. In total, the hit rates for TranSiGen_DISEASE and TranSiGen_DRUG are 38% and 80%, respectively. Here, a hit is defined as having an IC₅₀ less than 10 μM, a criterion comparable to that of the positive controls. The detailed prediction and experimental data for these top-ranking compounds are presented in Supplementary Table 12 and 13.

Specifically, among the active molecules identified by TranSiGen_DISEASE, HDAC inhibitors (panobinostat, praminostat, belinostat, apicidin, dacinostat, romidepsin, trichostatin A, alteminostat, mocetinostat and PCI-24781) consistently exhibit robust activities, indicating the promise of epigenomic therapeutics in pancreatic cancer³⁵. On the other hand, the dominant category among the top 50 comprises selective kinase inhibitors, including a recently discovered JAK kinase selective inhibitor (JAK2-IN-7, identified in 2020), the aurora kinase inhibitor AT9283, and the FLT3 inhibitor FLT3-IN-3. Their kinase inhibitory activities may serve as the underlying mechanism for their anti-YAPC effects³⁶. Additionally, we examined two Wnt/β-catenin inhibitors, hexachlorophene and tegatrabetan. While hexachlorophene has been reported to reduce the proliferation of pancreatic cells³⁷, our newly identified compound tegatrabetan (64.04 nM) exhibits significantly higher activity compared to hexachlorophene (8.83 μM).

.....

Fig. 6 Phenotype-based drug repurposing for the treatment of pancreatic cancer. **a** The flow chart of drug repurposing strategy. **b** The screening performance of phenotype-based strategy and structural similarity-based strategy. **c** TranSiGen_DISEASE screened compounds that are capable of inhibiting pancreatic cancer cells, and their max structural similarities to approved drugs. **d** The rankings of thiostrepton and resibufogenin from different screening strategies. **e** Top 50 compounds screened by TranSiGen_DISEASE and their respective cell proliferation inhibition activities. **f** Top 50 compounds screened by TranSiGen_DRUG and their corresponding cell proliferation inhibition activities.

Reviewer #3 (Remarks to the Author):

In the current manuscript authors have reported a novel Deep Learning based tool called TranSiGen to infer the chemical-induced transcriptional profile of small molecules with a potential for its application in the ligand-based screening, drug response prediction and phenotypic screening.

The methodology used is sound and the results support the outcome of this research. However, some of the issues need to be resolved as follows.

Reply:

Thank you for your thoughtful assessment of our work. We have diligently revised the manuscript based on your valuable and constructive comments and suggestions.

1. Introduction: The biological activity of a compound depends on its cellular permeability and it should be discussed in the introduction. Likewise, few sentences on the structural representation of small molecules (ECFP) can be added to the introduction part.

Reply:

Thank you for your valuable feedback. We appreciate the suggestion to discuss cellular permeability and the structural representation of small molecules (ECFP).

Cellular permeability is discussed in the **Discussion** section to ensure a focused Introduction. The relevant descriptions are as follows.

Discussion

The field of drug discovery is undergoing a significant transformation with the integration of artificial intelligence. While target-based approaches have been the dominant strategy, they are often hampered by limitations: (1) Many diseases lack clear and accessible protein targets, making it difficult to design effective drugs. (2) Targeting specific proteins can inadvertently interact with other molecules in the cell, leading to off-target effects and unexpected side effects. (3) Even when a target is identified, the resulting drug may struggle to reach its target within the cell due to poor cell permeability, thereby hindering the achievement of the desired therapeutic effect^{38,39}. These challenges have spurred the emergence of phenotype-based approaches, which directly analyze overall cellular response to drugs, offering a more holistic understanding of disease mechanisms, and the potential for discoveries of novel drug mechanisms and therapeutic opportunities^{13,40}.

.....

Regarding to structural representation of small molecules, we have added an explanation of ECFP to the “Molecular Representations” section of the **Methods** section, as follows:

Methods

Molecular Representations

.....

Alternatively, chemical fingerprints, widely used as a form of molecular representation in machine

learning, as it possesses the virtues of being lightweight, computationally efficient, and ability to capture key molecular features⁴¹. Accordingly, we employed chemical fingerprints as the molecular input for TranSiGen. There are represented as binary vectors indicating the presence or absence of particular substructures in compounds. Specifically, the molecular fingerprint ECFP4⁴² with a radius of 2 and a length of 2048 was used here.

2. Results: Since ECFP is a 2D descriptor representing a molecules, however, the 3D structure can contribute more towards learning how a molecules is interacting with the receptor or its target. What was the rationale behind using ECFP and can use of a 3D descriptor or ligand structure representation make the model more predictive?

Reply:

Thank you for your constructive comments. Given that each molecular representation has its own distinct advantages, the optimal choice of representation depends on the specific characteristics of the task at hand.

ECFP stands out as one of the most widely utilized 2D chemical fingerprints. It possesses the virtues of being lightweight, computationally efficient, and adept at capturing the salient features of molecules [*WIREs Comput. Mol. Sci.* **12**, e1603 (2022)]. Prior research has demonstrated that models based on 2D fingerprints perform comparably to state-of-the-art 3D structure-based models when predicting toxicity, solubility, partition coefficients, and protein-ligand binding affinity, when relying solely on ligand information [*Phys. Chem. Chem. Phys.* **22**, 8373–8390 (2020)].

3D-based representations excel in capturing crucial three-dimensional information about molecules, particularly in scenarios involving protein-ligand binding affinity with protein structures [*Phys. Chem. Chem. Phys.* **22**, 8373–8390 (2020)]. However, the acquisition of 3D descriptors is time-consuming due to the necessity of performing 3D conformation generation, evaluating, and finalizing the optimized conformation of molecules before calculating 3D descriptors.

Consequently, our primary focus lies in comparing the performance of TranSiGen-derived representation with the 2D fingerprint ECFP4 and the state-of-the-art pre-trained molecular representation KPGT [*Nat. Commun.* **14**, 7568 (2023)] in downstream tasks related to ligand-based virtual screening.

Additionally, in response to your suggestion, we incorporated a random forest (RF) model utilizing 3D descriptors as ligand structure representation for the ligand-based virtual screening task. The 3D descriptors for all molecules were computed using Mordred [*J. Cheminformatics* **10**, 4 (2018)], and the resulting 213-dimensional representations (Descriptors_3D) were employed as input features for the model. **Fig. R9** illustrates the ligand-based virtual screening performance of active compounds targeting HTR2A at various maximal similarity thresholds. Notably, for compounds dissimilar to the training set (chemical structure similarity $\in (0.0, 0.3]$), the TranSiGen-based model exhibits superior predictive capability compared to ECFP4-based, Descriptors_3D-based, and KPGT-based models. For relatively high structural similarity compounds (chemical structure similarity $\in (0.3, 0.1]$), the performance of TranSiGen-derived representation is comparable to both ECFP4-based and Descriptors_3D-based models. Similar trends in the scenario of dissimilar compounds are observed in ligand-based virtual screening for four additional evaluated targets (**Fig.**

R10). These findings further support the notion that leveraging transcriptional profiling, such as TranSiGen-derived representation, may confer advantages over structure-based representations in screening for novel scaffold compounds that deviate from known compound structures.

Fig. R9 Model performance of ligand-based virtual screening on target HTR2A within different thresholds of max similarity of test molecules relative to train data.

Fig. R10 Model performance of ligand-based virtual screening on other four target (DRD2, ADRA2A, SLC6A4 and KCNH2) within different thresholds of max similarity of test molecules relative to train data.

3. Methods: Overall the used methods are sound. Usage of ECFP as molecular representation can be justified.

Reply:

Thanks for your constructive comment. We added some descriptions of molecular fingerprint ECFP4 in section “Molecular Representations” in the **Methods**. The relevant descriptions are as follows.

Methods

Molecular Representations

.....

Alternatively, chemical fingerprints, widely used as a form of molecular representation in machine learning, as it possesses the virtues of being lightweight, computationally efficient, and ability to capture key molecular features⁴¹. Accordingly, we employed chemical fingerprints as the molecular input for **TranSiGen**. There are represented as binary vectors indicating the presence or absence of particular substructures in compounds. Specifically, the molecular fingerprint ECFP4⁴² with a radius of 2 and a length of 2048 was used here.

4. Overall the work described in this manuscript is sound and useful to the researchers working in the field of bioinformatics and drug discovery.

Reply:

Once again, we appreciate the time and effort you dedicated to reviewing our work.

Reviewer #1 (Remarks to the Author):

Thank authors for revisions and detailed responses. However, significant concerns persist without resolution on a rigorous basis.

TranSiGen's merit lies in its ability to predict chemical-induced transcriptomics for novel cell types. Otherwise, it is just an incremental improvement over existing methods. Nonetheless, TranSiGen's performance on the novel cell types remains inadequately assessed.

1. The random data split depicted in Figures 3 and 4 raises concerns. TranSiGen ought to undergo evaluation using a data split where the cell lines in the testing set differ significantly from those in the training and validation sets in terms of transcriptomic profiles.

2. MultiDCP is the sole method capable of predicting transcriptomic profiles for novel cell types among the chosen baseline models and is the most close to TranSiGen. Its performance should thus be depicted in Figures 3 and 4.

3. While FAME is intended for molecular design utilizing transcriptomic profiles, it employs contrastive learning for L1000 gene expression denoising representation. One of the manuscript's claimed contributions is this denoising representation. To substantiate this claim, authors should compare their denoising method with the contrastive learning method employed in FAME.

Reviewer #1 (Remarks on code availability):

I have tested the code and fixed the trivial errors but there are several ones left that prevented me from getting the training going, mostly because of the missing data path, although the author provided some of them in the Github repo:

1. Traceback (most recent call last):

```
File "~/TranSiGen/src/train_TrainSiGen_full_data.py", line 230, in <module>
train_TrainSiGen(args)
File "~/TranSiGen/src/train_TrainSiGen_full_data.py", line 115, in train_TrainSiGen
train = TranSiGenDataset(
File "~/TranSiGen/src/dataset.py", line 21, in __init__
with open('./data/LINCS2020/KPGT_emb2304.pickle', 'rb') as f:
FileNotFoundError: [Errno 2] No such file or directory: './data/LINCS2020/KPGT_emb2304.pickle'
```

I have tried the KPGT_emb2304_example.pickle (assuming this is the sample data), but I got error about the embedding cannot be found as below:

: KeyError: Caught KeyError in DataLoader worker process 0.

Original Traceback (most recent call last):

```
File "~/anaconda3/envs/deepMPI/lib/python3.8/site-packages/torch/utils/data/_utils/worker.py",
line 287, in _worker_loop
data = fetcher.fetch(index)
File "~/anaconda3/envs/deepMPI/lib/python3.8/site-packages/torch/utils/data/_utils/fetch.py",
line 49, in fetch
data = [self.dataset[idx] for idx in possibly_batched_index]
File "~/anaconda3/envs/deepMPI/lib/python3.8/site-packages/torch/utils/data/_utils/fetch.py",
line 49, in
data = [self.dataset[idx] for idx in possibly_batched_index]
File "~/TranSiGen/src/dataset.py", line 27, in __getitem__
mol_feature = self.smi2emb[self.idx2smi[self.mol_id[index]]]
KeyError: 'CN(C)S(=O)(=O)c1ccc2Sc3ccccc3N(CCCN3CCN(C)CC3)c2c1'
```

2. Cannot load the pretrained model. I chose random (training from scratch) but still getting the 1. error.

Traceback (most recent call last):

File "~/TranSiGen/src/train_TrainSiGen_full_data.py", line 230, in <module>

train_TrainSiGen(args)

File "~/TranSiGen/src/train_TrainSiGen_full_data.py", line 152, in train_TrainSiGen

model_base_x1 = torch.load(filename, map_location='cpu')

File "~/anaconda3/envs/deepMPI/lib/python3.8/site-packages/torch/serialization.py", line 608, in load

return _legacy_load(opened_file, map_location, pickle_module, **pickle_load_args)

File "~/anaconda3/envs/deepMPI/lib/python3.8/site-packages/torch/serialization.py", line 787, in _legacy_load

result = unpickler.load()

ModuleNotFoundError: No module named 'vae_x1'

Reviewer #2 (Remarks to the Author):

I have reviewed the new version of the manuscript and consider that the authors have satisfactorily addressed the reviewers' comments. The manuscript has improved its quality. Also, its contribution to the state of the art is clear.

Reviewer #3 (Remarks to the Author):

In response to the reviewer's comments, authors have now included new data and responded to most of the comments.

Reviewer #3 (Remarks on code availability):

The code at the given link looks fine.

REVIEWER COMMENTS

Reviewer #1 (Remarks to the Author):

Thank authors for revisions and detailed responses. However, significant concerns persist without resolution on a rigorous basis.

TranSiGen's merit lies in its ability to predict chemical-induced transcriptomics for novel cell types. Otherwise, it is just an incremental improvement over existing methods. Nonetheless, TranSiGen's performance on the novel cell types remains inadequately assessed.

Reply:

We sincerely thank the reviewer for thoroughly examining our manuscript and providing very helpful comments to guide our revision. We have revised the manuscript according to your suggestions to further demonstrate TranSiGen's performance in predicting chemical-induced transcriptomics for unseen cell types, and its effectiveness in downstream tasks. These revisions aim to further substantiate the merit of our work and reinforce the claims presented in the manuscript.

1. The random data split depicted in Figures 3 and 4 raises concerns. TranSiGen ought to undergo evaluation using a data split where the cell lines in the testing set differ significantly from those in the training and validation sets in terms of transcriptomic profiles.

Reply:

Thanks for your constructive comment. To achieve the goal that cell lines in the testing set differ significantly in transcriptomic profiles from those in the training and validation sets, we employed the challenging experimental setup proposed in MultiDCP [*PLoS Comput. Biol.* **18**, e1010367 (2022)] to assess the efficacy of models for unseen cell types.

For Figure 3, we conducted a comparative analysis between TranSiGen and MultiDCP, as well as DeepCE and CIGER for the cell-blind splitting scenario. Fig. 3c illustrates TranSiGen consistently outperforms the other models across various metrics including average Pearson's correlation coefficients (PCC), Positive Precision@100, and Negative Precision@100.

For Figure 4, we implemented downstream task, ligand-based virtual screening, following the above cell-blind splitting. The predicted perturbational profiles in cell-blind testing set were used as molecular representation for screening HTR2A active compounds. Fig. 4b shows that the model based on TranSiGen-derived representation outperforms other perturbational representations (DeepCE, CIGER and MultiDCP) by a significant margin in this cell-blind setting.

Fig. 3 The diagram of data splitting and the performance of inferring DEGs in different scenarios. **a** The diagram of chemical-blind splitting and cell-blind splitting. In scenario 1-1, a dataset of 355 compounds on 7 cell lines is split by compounds, ensuring that test compounds do not seen in the training set. In scenario 1-2, a complete dataset of 8,316 compounds on 164 cell lines is split by compounds. In scenario 2-2, the complete dataset of 8,316 compounds on 164 cell lines is split by cell lines. The model was trained using the profiling data of 10, 50, and 150 cell lines, and the prediction performance was evaluated on 7 new cell lines. **b** Model performance comparison in chemical-blind splitting. Error bars represent mean \pm SD. **c** Model performance comparison in cell-blind splitting (scenario 2-1). Error bars represent mean \pm SD. **d** The performance of TranSiGen in cell-blind splitting (scenario 2-2) by using different numbers of cell lines in the training set. Error bars represent mean \pm SD. Statistical t-test was applied between the models. (Note: ****, $p < 0.0001$; ***, $0.0001 < p \leq 0.001$; **, $0.001 < p \leq 0.01$; *, $0.01 < p \leq 0.05$ and ns, $0.05 < p \leq 1.0$)

Fig. 4 Model performance of ligand-based virtual screening on target HTR2A. **a** Performance of active compound prediction using different perturbational representations (chemical-blind). **b** Performance of active compound prediction using different perturbational representations (cell-blind). **c** Dimensionality reduction visualization of HTR2A active and inactive compounds based on various inferred perturbational representations. **d** Performance of active compound prediction by applying early fusion and late fusion for TranSiGen-derived representation from seven different cell lines. Error bars represent mean \pm SD. **e** Performance of active compounds prediction within different thresholds of max similarity of test molecules relative to train data. Error bars represent mean \pm SD. Statistical t-test was applied between the models. (Note: ****, $p < 0.0001$; ***, $0.0001 < p \leq 0.001$; **, $0.001 < p \leq 0.01$; *, $0.01 < p \leq 0.05$ and ns, $0.05 < p \leq 1.0$)

The related description in the revised manuscript are follows:

.....

Comparison with existing models in inferring differential expression genes

This section evaluates TranSiGen's performance in predicting DEGs compared to established baseline models. We benchmarked TranSiGen against DLEPS¹³, DeepCE¹⁴, CIGER¹⁵ and MultiDCP¹⁶. Notably, DLEPS, DeepCE, and CIGER primarily focus on *de novo* chemical profiling, with DeepCE and CIGER employing one-hot encoding for cell type distinction. In contrast, MultiDCP is the sole method among these baseline models that considers cellular context and specializes in predicting perturbational profiles for novel cell lines. Consequently, we assessed TranSiGen's performance in two settings:

(1) Chemical-blind splitting. This scenario includes two tests (Fig. 3a). In scenario 1-1, TranSiGen is compared to all models for its ability to predict DEGs of new compounds, using a dataset with 355 compounds across 7 cells to ensure comparability among models. In scenario 1-2, the complete dataset with 8,316 compounds across 164 cells is used to evaluate TranSiGen's scalability.

(2) Cell-blind splitting. This scenario also encompasses two tests. In scenario 2-1, we followed the challenging experimental setup proposed in MultiDCP¹⁶, where testing cell lines significantly differ from the training set, to predict DEGs for new cell lines. TranSiGen is compared to all models, excluding DLEPS due to its inability to distinguish cell types. In scenario 2-2 (Fig. 3a), TranSiGen is trained on 10, 50, and 150 cells, then evaluated on 7 new cells, to assess the benefit of expanding training cell types.

Results for chemical-blind splitting (scenario 1) are shown in Fig. 3b. Detailed results, including time consumption, computational resources, and metric scores, are provided in Supplementary Tables 1-3. TranSiGen excels in predicting DEGs for unseen compounds (scenario 1-1, Fig. 3b top). Compared to other models (DLEPS, DeepCE, CIGER and MultiDCP), it achieves a higher average PCC across seven cell lines. Additionally, TranSiGen outperforms these models in Positive Precision@100 and Negative Precision@100, metrics focusing on the most significantly regulated genes. Notably, TranSiGen's computational cost remains comparable. Furthermore, training on the complete dataset (scenario 1-2, Fig. 3b bottom) yields state-of-the-art performance. This significant improvement across seven cell lines compared to scenario 1-1 (Fig. 3b top) highlights TranSiGen's ability to leverage more training data for accurate DEG inference.

Results for cell-blind splitting (scenario 2) are shown in Fig. 3c and Fig. 3d. Comprehensive cross-validation results in Supplementary Table 4. When comparing TranSiGen with other models (excluding DLEPS) using the challenging experimental setup from MultiDCP, TranSiGen consistently outperforms other models in average PCC, Positive Precision@100, and Negative Precision@100 (scenario 2-1, Fig. 3c). Furthermore, TranSiGen's performance in inferring DEGs for unseen cell lines improves as the number of training cells increases (scenario 2-2, Fig. 3d and Supplementary Table 5).

In addition, we further explored the impact of different molecular representations and model initialization methods in the context of chemical-blind splitting. Initializing TranSiGen with perturbational profiles generated by gene knockdown yields superior performance compared to random initialization. Additionally, using pre-training representation, Knowledge-guided Pre-training of Graph Transformer (KPGT)²¹, further enhances the performance of inferring DEGs, surpassing the molecular fingerprint ECFP4 (as detailed in the "Molecular representations" in Method and corroborated by the metric scores in Supplementary Table 2 and 3).

Overall, these analyses underscore the efficacy of TranSiGen's self-supervised representation learning approach for transcriptional profiling. TranSiGen surpasses all baseline models in predicting DEGs for unseen compounds (chemical-blind splitting) and unseen cell lines (cell-blind splitting). Notably, despite the challenges of cross-cell prediction due to combined cell type and state influence on transcriptional profiles, TranSiGen demonstrates superior generalizability across cell lines. This suggests that TranSiGen effectively leverages basal cell profiles, potentially mitigating the impact of cell type.

Ligand-based virtual screening with TranSiGen-derived representation

Given that compounds with shared mechanisms induce similar gene expression profiles^{1,7,23},

we investigated the potential of predicted DEGs from TranSiGen as molecular representation for ligand-based virtual screening. First, as a proof-of-concept, Supplementary Fig. 3 demonstrates that active compounds targeting the same protein exhibit higher PCC compared to active and inactive ones. Subsequently, the TranSiGen-derived representation was used to assess whether a compound is active against a specific target. Specifically, we gathered and analyzed bioactivity data for compounds in LINCS2020¹⁹ from Pubchem²⁴ (refer to “Ligand-based virtual screening” in the Methods section). We identified five distinct targets, namely HTR2A, DRD2, ADRA2A, SLC6A4, and KCNH2, each having a significant number of active compounds, as illustrated in Supplementary Fig. 4. For each target, random forest (RF) classifiers were trained to differentiate active and inactive compounds. Notably, we evaluated the performance of predicted DEGs from TranSiGen along with other baseline models in both chemical-blind and cell-blind settings for these screening models.

Fig. 4a illustrates the performance of screening HTR2A (5-hydroxytryptamine receptor 2A) active compounds in chemical-blind setting, whereas Fig. 4b showcases the performance following the cell-blind setting. The model based on TranSiGen-derived representation outperforms other perturbational representations by a significant margin (Supplementary Table 6 and Supplementary Table 7). This result is further supported by the dimensionality reduction distribution of active/inactive compounds, where TranSiGen-derived representations clearly distinguish between the two, while other perturbational representations exhibit overlapped distributions (Fig. 4c and Supplementary Fig. 5).

.....

2. MultiDCP is the sole method capable of predicting transcriptomic profiles for novel cell types among the chosen baseline models and is the most close to TranSiGen. Its performance should thus be depicted in Figures 3 and 4.

Reply:

Thank you for your insightful comment. The performance comparison between TranSiGen and MultiDCP have been added in Figures 3 and 4. The detail of the implementation refers to the reply for question 1.

Fig. 3c shows the performance of inferring DEGs in cell-blind scenario, where TranSiGen performs better than MultiDCP, as well as DeepCE and CIGER.

Fig. 4b shows the performance of ligand-based virtual screening in cell-blind scenario. The HTR2A screening model based on TranSiGen-derived representation is significantly better than that based on MultiDCP, as well as DeepCE and CIGER.

3. While FAME is intended for molecular design utilizing transcriptomic profiles, it employs contrastive learning for L1000 gene expression denoising representation. One of the manuscript's claimed contributions is this denoising representation. To substantiate this claim, authors should compare their denoising method with the contrastive learning method employed in FAME.

Reply:

Thank you for your insightful recommendation. We have conducted an assessment to compare the efficacy of denoising representations from both FAME and TranSiGen in characterizing cellular features. Additionally, we evaluated the performance of these representations in various downstream tasks.

To ensure a fair comparison with FAME, we retrained TranSiGen using only two cell lines (MCF7 and PC3), aligning it with the dataset employed by FAME. As shown in Fig. R1, TranSiGen-derived representations effectively preserve cellular diversity, demonstrating a significant advantage in distinguishing different cell types compared to those derived from FAME.

Furthermore, TranSiGen's representations exhibit superior performance across a range of downstream tasks, including ligand-based virtual screening and drug response prediction (Table R1 and Table R2). Models leveraging TranSiGen-derived representation consistently outperform those based on FAME-derived representation, underscoring the superior profiling capabilities of TranSiGen.

Fig. R1 Dimensionality reduction visualization using denoising representations from TranSiGen and FAME.

Table R1. Model performance of ligand-based virtual screening on target KCNH2 using different perturbational representations.

Model	AUROC	AUPR	BACC	F1	Log_loss	MCC
FAME	0.909±0.063	0.709±0.218	0.736±0.051	0.545±0.091	0.321±0.064	0.431±0.116
TranSiGen	0.944±0.022	0.769±0.186	0.750±0.042	0.604±0.089	0.363±0.006	0.522±0.103

The bold font signifies the optimal performance.

Table R2. Model performance of drug response prediction.

Model	RMSE	Pearson	R ²
FAME	3.375±0.216	0.343±0.105	0.001±0.128
TranSiGen	2.947±0.094	0.518±0.061	0.240±0.048

The bold font signifies the optimal performance.

Reviewer #1 (Remarks on code availability):

I have tested the code and fixed the trivial errors but there are several ones left that prevented me from getting the training going, mostly because of the missing data path, although the author provided some of them in the Github repo:

1. Traceback (most recent call last):

```
File "~/TranSiGen/src/train_TrainSiGen_full_data.py", line 230, in <module>
train_TrainSiGen(args)
File "~/TranSiGen/src/train_TrainSiGen_full_data.py", line 115, in train_TrainSiGen
train = TranSiGenDataset(
File "~/TranSiGen/src/dataset.py", line 21, in __init__
with open('./data/LINCS2020/KPGT_emb2304.pickle', 'rb') as f:
FileNotFoundError: [Errno 2] No such file or directory:
'./data/LINCS2020/KPGT_emb2304.pickle'
```

I have tried the KPGT_emb2304_example.pickle (assuming this is the sample data), but I got error about the embedding cannot be found as below:

: KeyError: Caught KeyError in DataLoader worker process 0.

Original Traceback (most recent call last):

```
File ~/anaconda3/envs/deepMPI/lib/python3.8/site-packages/torch/utils/data/_utils/worker.py",
line 287, in _worker_loop
data = fetcher.fetch(index)
File ~/anaconda3/envs/deepMPI/lib/python3.8/site-packages/torch/utils/data/_utils/fetch.py", line
49, in fetch
data = [self.dataset[idx] for idx in possibly_batched_index]
File ~/anaconda3/envs/deepMPI/lib/python3.8/site-packages/torch/utils/data/_utils/fetch.py", line
49, in
data = [self.dataset[idx] for idx in possibly_batched_index]
File ~/TranSiGen/src/dataset.py", line 27, in __getitem__
mol_feature = self.smi2emb[self.idx2smi[self.mol_id[index]]]
KeyError: 'CN(C)S(=O)(=O)c1ccc2Sc3ccccc3N(CCCN3CCN(C)CC3)c2c1'
```

Reply:

Thanks for your constructive feedback. We have further provided the file “data_example” encompassing the perturbational profiles alongside the representation obtained from KPGT, to ensure a seamless training experience of TranSiGen.

2. Cannot load the pretrained model. I chose random (training from scratch) but still getting the 1. error.

Traceback (most recent call last):

```
File "~/TranSiGen/src/train_TrainSiGen_full_data.py", line 230, in <module>
train_TrainSiGen(args)
File "~/TranSiGen/src/train_TrainSiGen_full_data.py", line 152, in train_TrainSiGen
model_base_x1 = torch.load(filename, map_location='cpu')
```

```
File "~/anaconda3/envs/deepMPI/lib/python3.8/site-packages/torch/serialization.py", line 608, in
load
return _legacy_load(opened_file, map_location, pickle_module, **pickle_load_args)
File "~/anaconda3/envs/deepMPI/lib/python3.8/site-packages/torch/serialization.py", line 787, in
_legacy_load
result = unpickler.load()
ModuleNotFoundError: No module named 'vae_x1'
```

Reply:

Thanks for your constructive comment. We apologize for the oversight in the previously uploaded code, as it lacked the corresponding files. We have re-uploaded “vae_x1.py” and “vae_x2.py” to ensure the model can be trained smoothly.

Reviewer #2 (Remarks to the Author):

I have reviewed the new version of the manuscript and consider that the authors have satisfactorily addressed the reviewers' comments. The manuscript has improved its quality. Also, its contribution to the state of the art is clear.

Reply:

Thank you for your thorough review and positive feedback on the revised manuscript. We greatly appreciate your acknowledgment that we have adequately addressed the reviewers' comments and improved the quality of the manuscript.

Reviewer #3 (Remarks to the Author):

In response to the reviewer's comments, authors have now included new data and responded to most of the comments.

Reviewer #3 (Remarks on code availability):

The code at the given link looks fine.

Reply:

Thank you for your feedback. We appreciate the opportunity to enhance the manuscript and highly value your input in this process.

Reviewer #1 (Remarks to the Author):

All major concerns have been addressed. Thanks.

Reviewer #1 (Remarks on code availability):

The code is functional now.

REVIEWERS' COMMENTS

Reviewer #1 (Remarks to the Author):

All major concerns have been addressed. Thanks.

Reviewer #1 (Remarks on code availability):

The code is functional now.

Reply:

Thank you for your thorough review and positive feedback on the revised manuscript. We appreciate the opportunity to enhance the manuscript and highly value your input in this process.